# Resonant spin Hall effect of light in random photonic arrays

Federico Carlini[1,2] and Nicolas Cherroret[1]⋆

**1** Laboratoire Kastler Brossel, Sorbonne Université, CNRS,
ENS-PSL Research University, Collège de France, 4 Place Jussieu, 75005 Paris, France
**2** MajuLab, International Joint Research Unit UMI 3654, CNRS,
Université Côte d'Azur, Sorbonne Université, National University of Singapore,
Nanyang Technological University, Singapore

⋆ nicolas.cherroret@lkb.upmc.fr

## Abstract

It has been recently shown that the coherent component of light propagating in transversally disordered media, the so-called coherent mode, exhibits an optical spin Hall effect (SHE). In non-resonant materials, however, this phenomenon shows up at a spatial scale much larger than the mean free path, making its observation challenging due to the exponential attenuation of the coherent mode. Here, we show that in disordered photonic arrays exhibiting Mie resonances, the SHE on the contrary appears at a scale smaller than the mean free path if one operates in the close vicinity of the lowest transverse-magnetic resonance of the array. In combination with a weak measurement, this gives rise to a giant SHE that should be observable in optically-thin media. Furthermore, we show that by additionally exploiting the cooperative emission of a flash of light following the abrupt extinction of the incoming beam, one can achieve a time-dependent SHE, observable at large optical thickness as well.

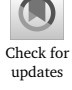

# 1 Introduction

The spin Hall effect (SHE) refers to a spin accumulation on the lateral surfaces of electric current carrying materials [1–4]. In contrast to the standard Hall effect, the SHE does not require any magnetic field but relies on the existence of a spin-orbit interaction. In the optical context, the SHE has a counterpart known as the SHE of light. Analogously to its electronic version, the SHE of light manifests itself by a transverse shift of right- or left-handed circularly-polarized beams in the plane perpendicular to the direction of propagation [5]. This effect typically arises as a result of the small coupling between the beam polarization and field gradient [6–8]. Spin Hall effects of light have been described in various optical systems, such as dielectric interfaces [9–11], gradient-index materials [12–14], non-paraxial beams [15–17], surface plasmonic systems [11,18,19] or for beams propagating along curved optical trajectories [20]. Although naturally small, such optical shifts are customarily measured by means of the technique of weak quantum measurements [9–11,21,22]. Furthermore, they are nowadays extensively studied in the context of structured materials such as photonic crystals, metasurfaces or metamaterials, whose peculiar properties can be exploited to enhance the shift by several orders of magnitudes [23–29].

Recently, it was shown that a SHE of light also exists in *disordered* structures, precisely in three-dimensional anisotropic random media displaying disorder in two directions only [30,31] ('transverse' disorder geometry [32–34]). In these systems, it was found that a SHE emerges in the coherent mode of the total optical signal transmitted through the medium. The coherent mode refers to the portion of light that remains coherent with the incident beam and, as such, propagates in the disorder without change in the direction of propagation [35]. In the case of a circularly-polarized beam of wavelength $\lambda$ impinging on the random medium at a finite angle of incidence $\theta$, the SHE corresponds to a lateral shift $\sim \lambda/(2\pi\sin\theta)$ of the centroid of the coherent mode, the effect being enhanced to the beam-width scale if a weak-measurement is performed [30]. In [31], it was also shown that a lateral shift arises for non-circularly-polarized beams as well, although with somewhat different properties.

A main obstacle to the detection of the SHE in a transversally disordered medium of thickness $L$, however, is the exponential attenuation of the coherent mode as $\exp(-L/\ell_{\mathrm{scat}})$, where $\ell_{\mathrm{scat}}$ is the scattering mean free path along the axis perpendicular to the

disordered plane. Indeed, without special care the SHE appears at a characteristic scale $\ell_S \sim \ell_{\text{scat}}/\sin^2\theta \gg \ell_{\text{scat}}$ where the coherent mode is no longer easily detectable. In [31], it was shown that $\ell_S$ could be reduced to a few mean free paths via a fine-tuning of the disorder correlation length. This strategy, however, may be inconvenient in practice as the disorder correlation is not necessarily controllable in structured materials. In this paper, we provide a decisive solution to this problem, by showing evidence of a SHE of light occuring at a scale smaller than the mean free path in transversally disordered photonic arrays exhibiting Mie resonances. Mie resonances correspond to strong enhancements in the scattering cross section of a single scatterer at specific frequencies. In ordered materials, they have been especially studied in relation with the band structure of photonic crystals [36,37], or in the context of dielectric metamaterials [38–40]. In disordered media, Mie resonances are also a precious tool to enhance scattering and wave-localization phenomena [41]. In the present work, we first show that tuning the optical frequency in the vicinity of the lowest transverse-magnetic (TM) Mie resonance of a transversally disordered array leads to a strong decrease of the spin Hall mean free path $\ell_S$ to a value much smaller than $\ell_{\text{scat}}$. In combination with a weak-measurement scheme, this gives rise to a giant SHE of the coherent mode, visible in optically-thin media ($L \sim \ell_{\text{scat}}$). Second, we provide a *temporal* description of the SHE following the abrupt extinction of the beam impinging on the photonic array. This protocol is known to generate a coherent flash of light, associated with the long dwell time of the resonant scatterers [42–44]. By again operating in the close vicinity of the TM resonance, we find that the coherent flash acquires a long-lived temporal tail, while a sizeable, time-dependent SHE emerges over the same time scale. This complementary setup thus offers the possibility to measure the SHE of light in optical-thick random media ($L \gg \ell_{\text{scat}}$) as well.

The paper is organized as follows. In Sec. 2, we first recall the main elements of light scattering by a long dielectric tube, the building block of our photonic array. This allows us to derive a simple analytical expression for the $T-$matrix of a single tube in the low-frequency limit, where only the two lowest Mie resonances are considered. We show, in particular, that the second resonance, addressed using a TM excitation, has a very narrow spectral width scaling as $\theta^2$. The core subject of the paper, the scattering of light by a transversally-disordered photonic array, is then introduced in Sec. 3. There we compute the transmission coefficient of the array and obtain an analytical formula for the disorder-average transmitted electric field. These results are exploited in Sec. 4, where we compute the SHE of the coherent mode and study its behavior in the vicinity of the TM resonance. In combination with a weak-measurement approach, we show that illuminating the array near this resonance leads to a giant SHE arising at a scale smaller than the mean free path. Sec. 5, finally, addresses the temporal dependence of SHE of light in a random array following the abrupt extinction of the incident beam. In the vicinity of the TM resonance, we reveal the existence of a long-lived coherent flash of light, together with a sizeable SHE emerging at a shorter time scale. Our main results are summarized in Sec. 6, and technical results are collected in three appendices.

## 2 Scattering of light by a narrow cylinder

### 2.1 $T$-matrix

In this section, we recall the main elements of the scattering theory of light by a single, infinitely-long dielectric cylinder, see Fig. 1(a). Our goal here is to provide a simple expression for the $T$-matrix of such a scatterer, which will be used in the next section as the building block of the problem of light scattering by a random photonic array.

Let us consider an incident, monochromatic plane wave $\mathbf{E}_{\text{in}}(\boldsymbol{r}) = \boldsymbol{E}_{\text{in}} \exp(i\boldsymbol{k}_{\text{in}} \cdot \boldsymbol{r})$, whose

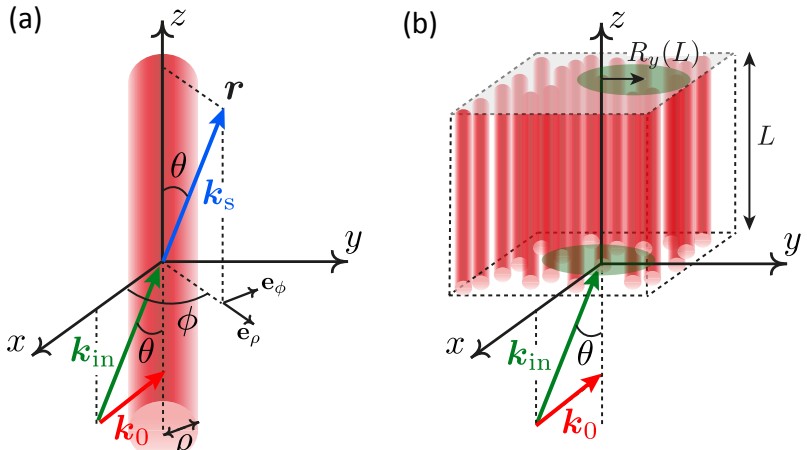

Figure 1: (a) Scattering of light by a single, infinitely-long cylinder of radius $\rho$. The incident wavevector $\boldsymbol{k}_{\text{in}}$ makes an angle $\theta$ with the cylinder axis $z$. During the scattering process, the $z-$component of the wavevector, $k_z = (\omega/c_0)\cos\theta$, is conserved. The projection of $\boldsymbol{k}_{\text{in}}$ in the transverse plane $(x, y)$ is denoted by $\boldsymbol{k}_0$. (b) Transmission of light through a random photonic array of dielectric cylinders (of thickness $L$). Upon crossing the array, the coherent mode intensity exhibits a spin Hall shift $R_y(L)$ along the $y-$axis.

wavevector $\boldsymbol{k}_{\text{in}} = \omega/c_0(-\sin\theta\boldsymbol{e}_x + \cos\theta\boldsymbol{e}_z)$ makes an angle $\theta$ with the cylinder axis $z$, see Fig. 1(a) ($\omega$ is the frequency and $c_0$ the vacuum speed of light). We express the constant field amplitude as $\boldsymbol{E}_{\text{in}} = E_i^{\text{TM}}\boldsymbol{e}_i^{\text{TM}} + E_i^{\text{TE}}\boldsymbol{e}_i^{\text{TE}}$, where the unit vectors $\boldsymbol{e}_i^{\text{TM}} = \cos\theta\boldsymbol{e}_x + \sin\theta\boldsymbol{e}_z$ and $\boldsymbol{e}_i^{\text{TE}} = -\boldsymbol{e}_y$ are such that $\boldsymbol{e}_i^{\text{TE}} \times \boldsymbol{e}_i^{\text{TM}} = \hat{\boldsymbol{k}}_{\text{in}}$ [45]. The cases $E_i^{\text{TM}} = 0$ and $E_i^{\text{TE}} = 0$ define the transverse-electric (TE) and transverse-magnetic (TM) modes, where the incident electric field has a zero (nonzero) component along the cylinder axis, respectively. In the presence of the cylinder, the total electric field decomposes as $\mathbf{E} = \mathbf{E}_{\text{in}} + \mathbf{E}_{\text{s}}$, where the scattered field $\mathbf{E}_{\text{s}}(\boldsymbol{r})$ at any point $\boldsymbol{r} = [\boldsymbol{r}_\perp = (x, y), z]$ is given by

$$\mathbf{E}_{\text{s}}(\boldsymbol{r}) = \int \frac{d^3\boldsymbol{k}'}{(2\pi)^3} e^{i\boldsymbol{k}'\cdot\boldsymbol{r}} \mathbf{G}_0(\boldsymbol{k}') \cdot \mathbf{T}(\boldsymbol{k}' - \boldsymbol{k}_{\text{in}}) \cdot \boldsymbol{E}_{\text{in}}. \tag{1}$$

Here $\mathbf{T}$ is the $T$-matrix of the cylinder, i.e. the scattering amplitude for the elastic process $\boldsymbol{k}_{\text{in}} \to \boldsymbol{k}'$, and $\mathbf{G}_0$ is the free-space Green's tensor:

$$\mathbf{G}_0(\boldsymbol{k}') = \left(\mathbb{I} - \frac{\boldsymbol{k}' \otimes \boldsymbol{k}'}{\omega^2/c_0^2}\right) \frac{1}{\omega^2/c_0^2 - \boldsymbol{k}'^2 + i0^+}, \tag{2}$$

where the $\otimes$ symbol refers to a dyadic product. A compact, diagrammatic representation of the scattered field (1) is represented in Fig. 2(a).

For a cylinder infinitely extended along $z$, the $z$-component of the wave vector is conserved during the scattering process. This imposes $\mathbf{T}(\boldsymbol{k}' - \boldsymbol{k}_{\text{in}}) = 2\pi\delta(k_z' - k_z)\mathbf{T}(\boldsymbol{k}_\perp' - \boldsymbol{k}_0)$, with $\boldsymbol{k}_\perp'$ and $\boldsymbol{k}_0 = -(\omega/c_0)\sin\theta\boldsymbol{e}_x$ respectively the projections of $\boldsymbol{k}'$ and $\boldsymbol{k}_{\text{in}}$ onto the transverse plane $(x, y)$, and $k_z'$ and $k_z = (\omega/c_0)\cos\theta$ their projections onto the $z$-axis. Equation (1) becomes:

$$\mathbf{E}_{\text{s}}(\boldsymbol{r}) = \left[\mathbb{I} - \frac{(k_z\boldsymbol{e}_z - i\boldsymbol{\nabla}_\perp) \otimes (k_z\boldsymbol{e}_z - i\boldsymbol{\nabla}_\perp)}{\omega^2/c_0^2}\right] \int \frac{d^2\boldsymbol{k}_\perp'}{(2\pi)^2} \frac{e^{i\boldsymbol{k}_\perp'\cdot\boldsymbol{r}_\perp + ik_z z}}{\omega^2/c_0^2 - k_z^2 - k_\perp'^2 + i0^+} \mathbf{T}(\boldsymbol{k}_\perp' - \boldsymbol{k}_0) \cdot \boldsymbol{E}_{\text{in}}. \tag{3}$$

Far from the cylinder axis, i.e. for $k_0 r_\perp \gg 1$, the angular integral is dominated by the stationary

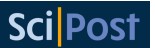

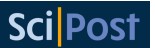

Figure 2: (a) Diagrammatic representation of the field (1) scattered by a single cylinder. The circled cross symbol represents the $T$-matrix of the cylinder. (b) Diagrammatic representation of the multiply scattered field (15). The sums run over the number $N$ of cylindrical scatterers. Remember that multiple scattering here takes place in the $(x, y)$ plane only.

phase corresponding to $\boldsymbol{k}'_\perp$ aligned with $\boldsymbol{r}_\perp$. The integral over $\boldsymbol{k}'_\perp$ then simplifies to

$$\int_0^\infty \frac{k'_\perp dk'_\perp}{(2\pi)^2} \sqrt{\frac{2\pi}{ik'_\perp r_\perp}} \frac{e^{ik'_\perp r_\perp + ik_z z}}{\omega^2/c_0^2 - k_z^2 - k_\perp'^2 + i0^+} \mathbf{T}(k'_\perp \boldsymbol{e}_\rho - \boldsymbol{k}_0) \cdot \boldsymbol{E}_{\text{in}}, \tag{4}$$

where $\boldsymbol{e}_\rho \equiv \boldsymbol{r}_\perp / r_\perp$. The remaining integral is computed by the residue theorem, yielding:

$$\mathbf{E}_s(\boldsymbol{r}) \simeq i \frac{e^{3i\pi/4}}{4} \sqrt{\frac{2}{\pi k_0 r_\perp}} e^{i\boldsymbol{k}_s \cdot \boldsymbol{r}} \left( \mathbb{I} - \hat{\boldsymbol{k}}_s \otimes \hat{\boldsymbol{k}}_s \right) \cdot \mathbf{T}(k_0 \boldsymbol{e}_\rho - \boldsymbol{k}_0) \cdot \boldsymbol{E}_{\text{in}}, \tag{5}$$

where $k_0 \equiv |\boldsymbol{k}_0| = \sqrt{\omega^2/c_0^2 - k_z^2}$ and we have introduced $\boldsymbol{k}_s \equiv k_0 \boldsymbol{e}_\rho + k_z \boldsymbol{e}_z$ with $\hat{\boldsymbol{k}}_s \equiv \boldsymbol{k}_s / k_s$, the wave vector of the scattered field at point $\boldsymbol{r}$, see Fig. 1(a). Note that due to transversality, $\hat{\boldsymbol{k}}_s \cdot \mathbf{E}_s = 0$. Therefore, the scattered field can naturally be decomposed along the vectors $\boldsymbol{e}_s^{\text{TM}} = -\cos\theta \boldsymbol{e}_\rho + \sin\theta \boldsymbol{e}_z$ and $\boldsymbol{e}_s^{\text{TE}} = \boldsymbol{e}_\phi$, so that $\boldsymbol{e}_s^{\text{TE}} \times \boldsymbol{e}_s^{\text{TM}} = \hat{\boldsymbol{k}}_s$. Denoting by $E_s^{\text{TM}}$ and $E_s^{\text{TE}}$ the two components of the electric field in this local basis, we can formally rewrite Eq. (5) as

$$\begin{bmatrix} E_s^{\text{TM}} \\ E_s^{\text{TE}} \end{bmatrix} = e^{3i\pi/4} \sqrt{\frac{2}{\pi k_0 r_\perp}} e^{i\boldsymbol{k}_s \cdot \boldsymbol{r}} \begin{bmatrix} T_1 & T_4 \\ T_3 & T_2 \end{bmatrix} \begin{bmatrix} E_i^{\text{TM}} \\ E_i^{\text{TE}} \end{bmatrix}. \tag{6}$$

The explicit expression of the coefficients $T_i$ can be calculated from Mie theory, by expanding the total field on the basis of vector cylindrical harmonics and imposing the continuity of tangential components of electric and magnetic fields at the cylinder boundary. The approach is a bit involved and can be found in [45]. It provides

$$T_1 = b_{0\text{I}} + 2 \sum_{n=1}^\infty b_{n\text{I}} \cos(n(\pi - \phi)),$$

$$T_2 = a_{0\text{II}} + 2 \sum_{n=1}^\infty a_{n\text{II}} \cos(n(\pi - \phi)), \tag{7}$$

$$T_3 = -2i \sum_{n=1}^\infty a_{n\text{I}} \sin(n(\pi - \phi)) = -T_4,$$

where $\pi - \phi$ is the angle between $\boldsymbol{k}_0$ and $k_0 \boldsymbol{e}_\rho$, see Fig. 1(a). The $a_n$ and $b_n$ are Mie coefficients that are functions of the size parameter $k\rho$, where $k \equiv \omega/c_0$ and $\rho$ is the radius of the cylinder. They exhibit an infinite set of resonances, see the example (9) below and Appendix A for their analytical expressions.

## 2.2 Low-frequency limit

In the rest of this manuscript, we assume a small angle of incidence, $\theta \ll 1$. We also focus on the low frequency limit, $k\rho \ll 1$. More precisely, we restrict ourselves to the two lowest Mie resonances of the problem, which turn out to constitute the minimal resonant model to capture the spin Hall effect (see Sec. 3). The lowest resonance is included in the Mie coefficients of order n = 1, which further satisfy $a_{1\text{II}} \simeq b_{1\text{I}} \simeq -ia_{1\text{I}}$ at low frequency and small angle, while the next one is in the coefficient $b_{0\text{I}}$. These observations allow us to express the $T$-matrix in terms of the coefficients $b_{0\text{I}}$ and $a_{1\text{I}}$ only:

$$\mathbf{T} = -i \begin{bmatrix} T_1 & T_4 \\ T_3 & T_2 \end{bmatrix} \simeq -i \begin{bmatrix} b_{0\text{I}} + 2ia_{1\text{I}}\cos\phi & 2ia_{1\text{I}}\sin\phi \\ -2ia_{1\text{I}}\sin\phi & 2ia_{1\text{I}}\cos\phi \end{bmatrix}. \tag{8}$$

The exact expression of $b_{0\text{I}}$ is, for instance, given by (see [45] or Appendix A)

$$b_{0\text{I}} = \frac{m^2 k\rho \sin\theta J_1(\eta)J_0(k\rho\sin\theta) - \eta J_0(\eta)J_1(k\rho\sin\theta)}{m^2 k\rho \sin\theta J_1(\eta)H_0^{(1)}(k\rho\sin\theta) - \eta J_0(\eta)H_1^{(1)}(k\rho\sin\theta)}, \tag{9}$$

where $\eta = k\rho\sqrt{m^2 - \cos^2\theta}$, with $m$ the ratio of the refractive index of the cylinder to that of the surrounding medium. The ratio (9) still includes an infinite set of resonances as a function of $k\rho$. As we wish to focus on the lowest one only, we further simplify Eq. (9) by linearizing the various Bessel functions around the first zero of the denominator. This calculation is tedious but straightforward, and leads to the simple Lorentzian approximation:

$$b_{0\text{I}} \simeq \frac{i\Gamma_1/2}{\omega - \omega_1 + i\Gamma_1/2}, \tag{10}$$

where, at small $\theta$, the resonance frequency $\omega_1$ and its bandwidth $\Gamma_1$ are given by

$$\omega_1 \simeq \frac{c_0 \alpha}{\rho\sqrt{m^2 - 1}}, \quad \text{and} \quad \Gamma_1 \simeq \frac{c_0 m\alpha\pi \sin^2\theta}{\rho(m^2 - 1)}, \tag{11}$$

with $\alpha \simeq 2.4048$. Notice that $\Gamma_1(\theta \to 0) \propto \theta^2 \to 0$, a key property that will be at the origin of a giant spin Hall effect in the random array. A similar approximation for $a_{1\text{I}}$ gives:

$$a_{1\text{I}} \simeq -\frac{1}{2}\frac{\Gamma_0/2(\omega/\omega_0)^2}{\omega - \omega_0 + i\Gamma_0/2(\omega/\omega_0)^2}. \tag{12}$$

The resonance frequency $\omega_0$ and the bandwidth $\Gamma_0$ have more complicated expressions, which are given in Appendix B for clarity. They show, in particular, that in contrast to $\Gamma_1$, at small angle $\Gamma_0$ varies very weakly with $\theta$. Equations (8), (10) and (12) constitute a minimal model for light scattering by a cylinder at low frequency, where only the two lowest Mie resonances are considered.

## 2.3 Properties of lowest Mie resonances

To conclude this section, we briefly discuss the main properties of the two resonances (10) and (12), as well as the accuracy of the Lorentzian approximations made. To this aim, it is convenient to evaluate the scattering cross section $\sigma_s$ of the cylinder, defined as

$$\sigma_s = \frac{\int_A \mathbf{\Pi}_s . \mathbf{e}_\rho \, dA}{|\mathbf{\Pi}_i|}, \tag{13}$$

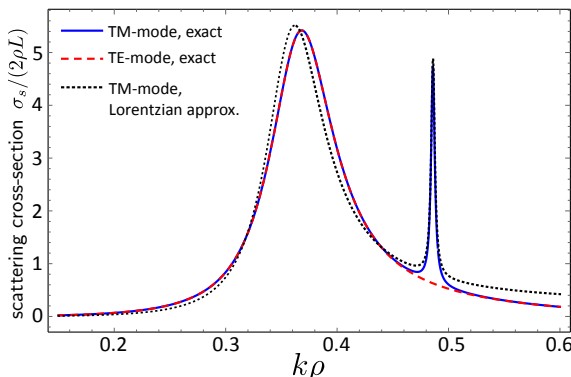

Figure 3: Scattering cross-section for the TE and TM modes, computed from Eq. (14) using $m = 5$ and $\sin\theta = 0.05$. Here we only show the two lowest Mie resonances. The solid and dashed curve are the result of the exact Mie calculation for the TM and TE modes, respectively. In the TM case, a very sharp resonance of width $\propto \sin^2\theta$ generically shows up at $\omega = \omega_1$. The dotted curve shows the Lorentzian approximation, Eqs. (10) and (12), in the TM mode for comparison.

where $\mathbf{\Pi}_{i,s} = 1/(2i\omega\mu_0)\Re[\mathbf{E}_{i,s} \times (\nabla \times \mathbf{E}_{i,s}^*)]$ is the Poynting vector of the incident and scattered fields, and $A$ is a fictitious surface enclosing the cylinder in the far field. Inserting Eq. (6) into this definition, we find

$$\frac{\sigma_s^{\text{TE}}}{2\rho L} = \frac{2}{k\rho}\langle T_2^2 + T_4^2 \rangle_\phi, \quad \frac{\sigma_s^{\text{TM}}}{2\rho L} = \frac{2}{k\rho}\langle T_1^2 + T_3^2 \rangle_\phi, \tag{14}$$

for the TE ($E_i^{\text{TM}} = 0$) and TM ($E_i^{\text{TE}} = 0$) polarization modes. In Eq. (14), $\langle \ldots \rangle_\phi$ refers to an angular average over $\phi$ and we have normalized $\sigma_s$ by the geometrical cross section $2\rho L$ of the cylinder. The scattering cross section in the two polarization modes is shown in Fig. 3. The prediction of Mie theory, based on the exact expressions (7) of the coefficients of the $T$-matrix, is displayed together with the Lorentzian approximation in the TM case, based on Eqs. (8), (10) and (12). The plot confirms the good accuracy of the simple Lorentzian model, on which we will rely in the rest of the paper.

Fig. 3 also confirms that the resonance at $\omega_0$ is the lowest one. This resonance shows up both in the TE and TM modes, and is rather broad even for large values of the relative refractive index $m$ of the cylinder. As mentioned above, both $\omega_0$ and $\Gamma_0$ weakly depend on the angle of incidence $\theta$. The second resonance, at $\omega_1 > \omega_0$, only shows up in the TM mode, and for this reason will be referred to as the 'TM resonance' in the following. Its most remarkable property is its sharpness, which stems from the proportionality of $\Gamma_1$ to $\sin^2\theta \ll 1$ at small $\theta$, see Eq. (11). In Sec. 4, this property will be at the basis of a giant spin Hall effect in random photonic arrays of cylinders excited in the close vicinity of $\omega_1$.

# 3 Scattering of light by a random photonic array

## 3.1 Dyson equation

Let us now consider scattering of light by an array made of a large number of identical, randomly distributed cylindrical scatterers with parallel axes, see Fig. 1(b). The electric field $\mathbf{E}(\mathbf{r})$ at some point $\mathbf{r}$ within the array is now given by a sum over multiple scattering trajectories, as sketched in Fig. 2(b). In terms of the Fourier components

$E(\mathbf{k}_\perp, k_z) = \int d^2\mathbf{r}_\perp dz e^{-i\mathbf{k}_\perp \cdot \mathbf{r}_\perp - i k_z z} E(\mathbf{r})$, the multiple scattering sequence is iterated by the relation

$$\mathbf{E}(\mathbf{k}_\perp, k_z) = \mathbf{E}_{\text{in}}(\mathbf{k}_\perp, k_z) + \sum_j \int \frac{d^2\mathbf{k}'_\perp}{(2\pi)^3} \mathbf{G}_0(\mathbf{k}'_\perp, k_z) \cdot \mathbf{T}_j(\mathbf{k}_\perp - \mathbf{k}'_\perp) \cdot \mathbf{E}(\mathbf{k}'_\perp, k_z), \qquad (15)$$

where $\mathbf{T}_j$ refers to the $T$-matrix of the cylinder $j$ and the sum runs over the number $N$ of cylinders of the array. This relation generalizes the single scattering solution (1). Notice the conservation of $k_z$ due to translation invariance along $z$. Denoting by $\mathbf{r}_{\perp j}$ the (random) position of the cylinder $j$ in the transverse plane $(x, y)$, we have $\mathbf{T}_j(\mathbf{k}_\perp - \mathbf{k}'_\perp) \simeq \mathbf{T} e^{-i(\mathbf{k}_\perp - \mathbf{k}'_\perp) \cdot \mathbf{r}_{\perp j}}$, where $\mathbf{T}$ is the $T$−matrix of the cylinder located at $\mathbf{r}_{\perp j} = 0$.

From now on, we focus on the statistical average $\overline{\mathbf{E}}$ of Eq. (15) over the cylinder positions, which we denote by an overbar. In general, this average can not be computed analytically exactly due to the multiple correlations between the scatterers. Here, however, we assume a dilute distribution of the cylinders, i.e. $n k_0^{-2} \ll 1$, with $n$ the surface density of the cylinders in the transverse plane $(x, y)$ and $k_0 = \omega/c_0 \sin\theta$ the transverse wave number. Together with the low-frequency limit introduced in Sec. 2.2, we thus have:

$$\rho \ll k_0^{-1} \ll n^{-1/2}. \qquad (16)$$

The diluteness condition $n k_0^{-2} \ll 1$ implies that the probability that the scattered light "returns" to a scatterer already visited is very small, such that we have $\overline{\mathbf{T}_j \cdot \mathbf{E}_s} \simeq \overline{\mathbf{T}_j} \cdot \overline{\mathbf{E}_s}$ (independent-scattering approximation). On the other hand, the condition $n\rho^2 \ll 1$ implies that spatial correlations between tubes, caused by their finite size, are negligible (see [46] for details about the case of correlated or partially disordered media). This allows us to employ a simple model of disorder where the cylinder positions $\mathbf{r}_{\perp j}$ are uniformly distributed, such that:

$$\sum_j \overline{e^{-i(\mathbf{k}_\perp - \mathbf{k}'_\perp) \cdot \mathbf{r}_{\perp j}}} \equiv \sum_j \int \frac{d^2\mathbf{r}_{\perp j}}{S} e^{-i(\mathbf{k}_\perp - \mathbf{k}'_\perp) \cdot \mathbf{r}_{\perp j}} \simeq \frac{N}{S} \delta(\mathbf{k}_\perp - \mathbf{k}'_\perp), \qquad (17)$$

with $N/S = n$, $S$ being the surface of the array in the transverse plane $(x, y)$. The disorder average of Eq. (15) thus reads

$$\overline{\mathbf{E}}(\mathbf{k}_\perp, k_z) = \mathbf{E}_{\text{in}}(\mathbf{k}_\perp, k_z) + \mathbf{G}_0(\mathbf{k}_\perp, k_z) \cdot n\mathbf{T} \cdot \overline{\mathbf{E}}(\mathbf{k}_\perp, k_z). \qquad (18)$$

For a point-source emitting light within the array, this relation is simply generalized in terms of a disorder-average Green's tensor $\overline{\mathbf{G}}$, which obeys

$$\overline{\mathbf{G}}(\mathbf{k}_\perp, k_z) = \mathbf{G}_0(\mathbf{k}_\perp, k_z) + \mathbf{G}_0(\mathbf{k}_\perp, k_z) \cdot n\mathbf{T} \cdot \overline{\mathbf{G}}(\mathbf{k}_\perp, k_z). \qquad (19)$$

Equation (19) is known as the Dyson equation at the independent-scattering approximation [41]. From the knowledge of the $T$-matrix of a single cylinder, the average Green's tensor of the random array thus follows from

$$\overline{\mathbf{G}}(\mathbf{k}_\perp, k_z) = [G_0^{-1}(\mathbf{k}_\perp, k_z) - \mathbf{\Sigma}]^{-1}, \qquad (20)$$

with $\mathbf{\Sigma} \equiv n\mathbf{T}$, a quantity known as the self-energy tensor [41].

## 3.2 Disorder-average transmitted field

In this paper, we aim at describing the coherent component of light transmitted through a random photonic array of thickness $L$, as illustrated in Fig. 1(b). This "coherent mode" refers to the portion of the total scattered signal that propagates in the forward direction through

the array, namely in the same direction as that of the incident beam. It is described by the disorder average of the transmitted field, whose Fourier distribution follows from the input-output relation

$$\overline{\mathbf{E}}(\boldsymbol{k}_\perp, z\!=\!L) = \mathbf{t}(\boldsymbol{k}_\perp, L) \cdot \mathbf{E}(\boldsymbol{k}_\perp, z\!=\!0), \tag{21}$$

where $\mathbf{E}(\boldsymbol{k}_\perp, z = 0)$ is the Fourier distribution of the field at the entrance of the array. In the following, we take the latter of the form

$$\mathbf{E}(\boldsymbol{k}_\perp, z\!=\!0) = \sqrt{2\pi} w_0 \exp[-(\boldsymbol{k}_\perp - \boldsymbol{k}_0)^2 w_0^2/4] \boldsymbol{e}_0, \tag{22}$$

with $k_0 w_0 \gg 1$. This models a collimated beam of intensity distribution normalized to unity, i.e. $\int d^2\boldsymbol{k}_\perp/(2\pi)^2 |\mathbf{E}(\boldsymbol{k}_\perp, z = 0)|^2 = 1$, and of unit polarization vector $\boldsymbol{e}_0$. A fundamental quantity in Eq. (21) is the average transmission matrix of the array $\mathbf{t}$. The latter is given by the Fisher-Lee formula [47]:

$$\mathbf{t}(\boldsymbol{k}_\perp, L) = 2i\sqrt{k^2 - \boldsymbol{k}_\perp^2}\, \overline{\mathbf{G}}(\boldsymbol{k}_\perp, z\!=\!L), \tag{23}$$

where $\overline{\mathbf{G}}(\boldsymbol{k}_\perp, z) \equiv \int dk_z/(2\pi) e^{ik_z z} \overline{\mathbf{G}}(\boldsymbol{k}_\perp, k_z)$.

From Eqs. (21) and (23), the problem thus reduces to evaluating the average Green's tensor of the array. This can be done by computing the tensor inverse in the right-hand side of the Dyson equation (20). This problem was previously tackled in [30,31], in the simpler case of a continuous, non-resonant random medium. To make contact with these works, it is convenient to rewrite the $T-$matrix (8) of a single scatterer in the global basis $(x, y, z)$. At small angle $\theta \ll 1$, we find after a basis change:

$$\boldsymbol{\Sigma} = \begin{bmatrix} \Sigma_{xx} & 0 & 0 \\ 0 & \Sigma_{xx} & 0 \\ 0 & 0 & \Sigma_{zz} \end{bmatrix}_{(x,y,z)}, \tag{24}$$

where

$$\Sigma_{xx} = -8n a_{1\mathrm{I}} = 4n \frac{\Gamma_0/2(\omega/\omega_0)^2}{\omega - \omega_0 + i\Gamma_0/2(\omega/\omega_0)^2}, \tag{25}$$

and

$$\Sigma_{zz} = -\frac{4in b_{0\mathrm{I}}}{\sin^2\theta} = \frac{4n}{\sin^2\theta}\frac{\Gamma_1/2}{\omega - \omega_1 + i\Gamma_1/2}. \tag{26}$$

Note that the fact that the self-energy tensor is diagonal is not related to the resonant character of the photonic array, but to its uniaxial anisotropy. In particular, the same diagonal structure was found in [30] within a non-resonant, continuous model of transverse disorder. Inserting Eq. (24) into Eq. (20) and using Eq. (23), we obtain [30,31]

$$\begin{aligned}
\mathbf{t}(\boldsymbol{k}_\perp, L) = e^{ik_z L}\Big[ &e^{-i\Sigma_{\mathrm{TE}}L/2k_z}\delta_{ij} - e^{-i\Sigma_{\mathrm{TM}}L/2k_z}\hat{k}_i\hat{k}_j \\
&+ \frac{e^{-i\Sigma_{\mathrm{TE}}L/2k_z} - e^{-i\Sigma_{\mathrm{TM}}L/2k_z}}{\hat{k}_\perp^2}\left(\delta_{iz}\hat{k}_j\hat{k}_z + \delta_{jz}\hat{k}_i\hat{k}_z - \delta_{iz}\delta_{jz} - \hat{k}_i\hat{k}_j\right) \Big],
\end{aligned} \tag{27}$$

where $i, j = x, y, z$ and $\hat{k}_i \equiv k_i/k$ with $k_z \equiv \sqrt{\omega^2/c^2 - \boldsymbol{k}_\perp^2}$. The two complex rates $\Sigma_{\mathrm{TE,TM}}$ are defined as

$$\Sigma_{\mathrm{TE}} = \Sigma_{xx}, \quad \Sigma_{\mathrm{TM}} = \Sigma_{xx} + (\Sigma_{zz} - \Sigma_{xx})\sin^2\theta, \tag{28}$$

where the 'TE' and 'TM' labeling will be shortly clarified. Combining Eqs. (21), (22), (23) and (27), we finally obtain the exact expression of the average transmitted field:

$$\overline{\mathbf{E}}(\boldsymbol{k}_\perp, L) = \sqrt{2\pi} w_0 e^{-(\boldsymbol{k}_\perp - \boldsymbol{k}_0)^2 w_0^2/4} e^{ik_z L}\Big[ e^{-i\Sigma_{\mathrm{TE}}L/2k_z}\boldsymbol{e}_0 + (e^{-i\Sigma_{\mathrm{TM}}L/2k_z} - e^{-i\Sigma_{\mathrm{TE}}L/2k_z})\boldsymbol{p}(\boldsymbol{k}_\perp) \Big]. \tag{29}$$

Here $p(k_\perp) \equiv (e_\perp \cdot e_0)e_\perp + (e_z \cdot e_0)e_z$ is the projection of the initial polarization $e_0$ onto the $(e_\perp \equiv k_\perp/k_\perp, e_z)$ plane.

Equation (29), together with Eqs. (25) and (26), will be at the basis of all subsequent results of the paper. The physics behind this equation, however, deserves a few comments. First, as is common in random media, Eq. (29) indicates that the average field is exponentially attenuated with $L$ (due to the imaginary part of the self-energies $\Sigma_{\text{TE,TM}}$). This attenuation fundamentally originates from the destructive interference between the incident beam and the light scattered in the forward direction $\hat{k}_0$ in the transverse plane [35]. Here, however, the attenuation involves *two* different self-energies $\Sigma_{\text{TE}}$ and $\Sigma_{\text{TM}}$, which turn out to correspond to the two polarization modes of the problem discussed in Sec. 2. Indeed, for $e_0 = e_i^{\text{TE,TM}}$ Eq. (29) reduces to $\overline{\mathbf{E}}(k_0, z) \propto e^{-i\Sigma_{\text{TE,TM}}z/2k_z} e_i^{\text{TE,TM}}$. Second, while the first term in the right-hand side of Eq. (29) describes an evolution of the average field without deformation of its spatial profile, the latter is modified by the second term, via a coupling between the transverse wave vector $k_\perp$ and the polarization $e_0$, encoded in $p(k_\perp)$. This showcases the phenomenon of spin-orbit interaction of light, which is responsible for the spin Hall effect addressed in the next section.

# 4 Resonant spin Hall effect

## 4.1 Intensity distribution of the coherent mode

We now evaluate the spatial, intensity distribution of the coherent mode of the photonic array, defined as $I(r_\perp, L) \equiv |\overline{\mathbf{E}}(r_\perp, L)|^2$, where $\overline{\mathbf{E}}(r_\perp, L)$ is the Fourier transform of Eq. (29). Introducing $k_\perp^\pm = k_\perp \pm q/2$, we can write

$$I(r_\perp, L) = \int \frac{d^2 k_\perp}{(2\pi)^2} \int \frac{d^2 q}{(2\pi)^2} e^{iq \cdot r_\perp} \overline{\mathbf{E}}(k_\perp^+, L) \cdot \overline{\mathbf{E}}^*(k_\perp^-, L). \tag{30}$$

The incident beam (22) being collimated around $k_\perp = k_0$, the product of the two average fields can be expanded at small $q$:

$$\overline{\mathbf{E}}(k_\perp^+, L) \cdot \overline{\mathbf{E}}^*(k_\perp^-, L) \simeq e^{-w_0^2 q^2/8} \{|\overline{\mathbf{E}}(k_\perp, L)|^2 + q \cdot \nabla_q[\overline{\mathbf{E}}(k_\perp^+, L) \cdot \overline{\mathbf{E}}^*(k_\perp^-, L)]|_{q \to 0}\}. \tag{31}$$

This leads to

$$I(r_\perp, L) = \int \frac{d^2 q}{(2\pi)^2} e^{-w_0^2 q^2/8} [1 - iq \cdot \mathbf{R}_\perp(L)] e^{iq \cdot r_\perp} \int \frac{d^2 k_\perp}{(2\pi)^2} |\overline{\mathbf{E}}(k_\perp, L)|^2, \tag{32}$$

where

$$\mathbf{R}_\perp(L) = \frac{i \int \frac{d^2 K}{(2\pi)^2} \nabla_q[\overline{\mathbf{E}}(K_+, L) \cdot \overline{\mathbf{E}}^*(K_-, L)]|_{q \to 0}}{\int \frac{d^2 K}{(2\pi)^2} |\overline{\mathbf{E}}(K, L)|^2} \equiv \frac{\int d^2 r_\perp r_\perp I(r_\perp, L)}{\int d r_\perp I(r_\perp, L)}, \tag{33}$$

is the beam centroid in the transverse plane $(x, y)$. The integral over $q$ in Eq. (32) is dominated by small $q$−values, so that we can accurately replace the term inside the squared brackets by an exponential factor. This finally gives:

$$I(r_\perp, L) \simeq I(L) e^{-2|r_\perp - \mathbf{R}_\perp(L)|^2/w_0^2}, \tag{34}$$

where

$$I(L) \equiv I_0 \int \frac{d^2 k_\perp}{(2\pi)^2} |\overline{\mathbf{E}}(k_\perp, L)|^2 = I(\mathbf{R}_\perp(L), L), \tag{35}$$

with $I_0 \equiv 2/(\pi w_0^2)$. Equation (34) describes a rigid, spatial shift $\mathbf{R}_\perp(L)$ of the coherent mode in the transverse plane $(x, y)$ at the output of the medium. We will see next that the most interesting effects due to the spin-orbit term in Eq. (29) manifest themselves through this shift.

## 4.2 Intensity at the beam center

We first discuss the intensity $I(L)$ of the coherent mode at the beam center, which can be computed from Eqs. (29) and (35). For TE- or TM-polarized light (i.e. $\boldsymbol{e}_0 = \boldsymbol{e}_i^{\text{TE}}$ or $\boldsymbol{e}_i^{\text{TM}}$), we find at small angle

$$I(L) = I_0 \exp\left(\frac{L\,\text{Im}\Sigma_{\text{TE,TM}}}{k}\right), \tag{36}$$

which describes an exponential decay respectively governed by the imaginary parts $\text{Im}\Sigma_{\text{TE}} < 0$ or $\text{Im}\Sigma_{\text{TM}} < 0$. As usually in a disordered medium, this decay provides an effective-medium description for the propagation of the coherent mode, which gets depleted at the scale of a mean free path due to scattering in other directions [35]. As mentioned above, a difference with standard isotropic random media is that the decay here involves two different mean free paths $-k/\text{Im}\Sigma_{\text{TE,TM}}$, one for each mode. In the following, we will use as a benchmark for all length scales along $z$ the TE scattering mean free path, denoted by $\ell_{\text{scat}}$:[1]

$$\frac{1}{\ell_{\text{scat}}} \equiv -\frac{1}{k}\text{Im}\,\Sigma_{\text{TE}} = \frac{4n}{k}\frac{(\Gamma_0/2)^2(\omega/\omega_0)^4}{(\omega - \omega_0)^2 + (\Gamma_0/2)^2(\omega/\omega_0)^4}, \tag{37}$$

where we have used Eqs. (25) and (28) in the second equality. We will also frequently use the value of the mean free path exactly at the resonance, $\ell_{\text{scat}}^0 \equiv \ell_{\text{scat}}(\omega_0) = k/4n$.

For an arbitrary polarization of the incident beam, $I(L)$ is a weighted sum of the two exponential decays (36). For a reason that will be discussed in the next section, the emergence of a nonzero spin Hall effect imposes the use of a laser populating both modes. In particular, for a balanced mixture of the two modes [see Eq. (43) below], we have:

$$I(L) = \frac{I_0}{2}\left(e^{L\,\text{Im}\Sigma_{\text{TE}}/k} + e^{L\,\text{Im}\Sigma_{\text{TM}}/k}\right). \tag{38}$$

It is instructive, at this stage, to display the coherent mode intensity (38) as a function of the frequency, see Fig. 4. The intensity exhibits two local minima, corresponding to the two resonances at $\omega_0$ and $\omega_1$ [with the TM resonance at $\omega_1$ originating from $\Sigma_{\text{TM}}$ only, see Eq. (28)]. The plot, in particular, showcases the sharp character of the TM resonance, whose width $\Gamma_1 \propto \sin^2\theta$.

## 4.3 Resonant spin Hall effect

We now examine the beam centroid $\mathbf{R}_\perp(L)$, which is the most important feature of Eq. (34). To this aim, we calculate the integrand in Eq. (33) using the general expression (29) of the momentum distribution of the average field. For an arbitrary polarization $\boldsymbol{e}_0$, this integrand

---

[1]For the balanced polarization mixture considered hereafter, the coherent intensity is given by Eq. (38), which also involves the TM mean free path $-\text{Im}\Sigma_{\text{TM}}/k$. The definition (37), nevertheless, turns to be a valid choice whatever the frequency. Indeed, as discussed in Sec. 4.3, away from the TM resonance $\text{Im}\Sigma_{\text{TM}} \simeq \text{Im}\Sigma_{\text{TE}}$, while at the TM resonance $|\text{Im}\Sigma_{\text{TM}}| \gg |\text{Im}\Sigma_{\text{TE}}|$. Therefore, in both cases $I(L) \propto \exp(L\text{Im}\Sigma_{\text{TE}}/k)$ at large enough $L$.

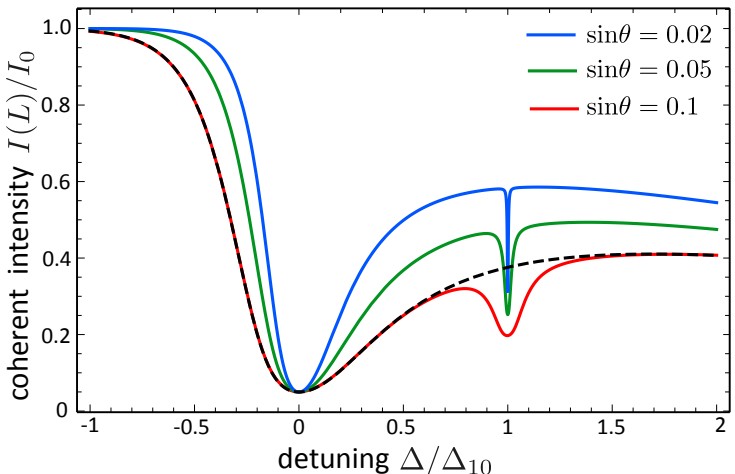

Figure 4: Intensity $I(L)$ of the coherent mode at the beam center vs. frequency, for a balanced mixture of the TE/TM modes [Eq. (38)] and for three values of the angle of incidence $\theta$. Here $\Delta = (\omega - \omega_0)/(\Gamma_0/2)$ is expressed in units of $\Delta_{10} = (\omega_1 - \omega_0)/(\Gamma_0/2)$. In each case, the sharp resonance at $\omega = \omega_1$ ($\Delta/\Delta_{10} = 1$) is visible, and becomes narrower as $\theta$ decreases. The relative cylinder refractive index is set to $m = 3$ and the optical thickness to $L = 3\ell_{\text{scat}}^0$. The dashed curve shows the intensity in the TE mode for comparison, i.e. $I(L) = I_0 \exp(-L/\ell_{\text{scat}})$.

reads:

$$\overline{\mathbf{E}}(\mathbf{k}_\perp^+, L) \cdot \overline{\mathbf{E}}^*(\mathbf{k}_\perp^-, L) \simeq 2\pi w_0^2 \exp\left[-w_0^2(\mathbf{k}_\perp - \mathbf{k}_0)^2/2 - w_0^2\mathbf{q}^2/8 + i(k_z^+ - k_z^-)L\right] \tag{39}$$
$$\times \left\{|e^{-i\Sigma_{\text{TE}}L/2k_z}|^2 - |\mathbf{e}_0 \cdot \mathbf{e}_\perp^-|^2\left[|e^{-i\Sigma_{\text{TE}}L/2k_z}|^2 - e^{-i(\Sigma_{\text{TE}} - \Sigma_{\text{TM}}^*)L/2k_z}\right]\right.$$
$$- |\mathbf{e}_0 \cdot \mathbf{e}_\perp^+|^2\left[|e^{-i\Sigma_{\text{TE}}L/2k_z}|^2 - e^{-i(\Sigma_{\text{TM}} - \Sigma_{\text{TE}}^*)L/2k_z}\right]$$
$$+ (\mathbf{e}_0 \cdot \mathbf{e}_\perp^+)(\mathbf{e}_0 \cdot \mathbf{e}_\perp^-)\left[|e^{-i\Sigma_{\text{TE}}L/2k_z}|^2 + |e^{-i\Sigma_{\text{TM}}L/2k_z}|^2 - e^{-i(\Sigma_{\text{TE}} - \Sigma_{\text{TM}}^*)L/2k_z} - e^{-i(\Sigma_{\text{TM}} - \Sigma_{\text{TE}}^*)L/2k_z}\right]\right\},$$

where $\mathbf{e}_\perp^\pm = \mathbf{e}_\perp \pm \mathbf{q}/(2k_\perp)$ and $k_z^\pm = \sqrt{k^2 - \mathbf{k}_\perp^\pm}$. In this expression, the non-zero contributions to the $\mathbf{q}$-gradient in Eq. (33) are

$$\nabla_{\mathbf{q}} \exp\left[i(k_z^+ - k_z^-)L\right]|_{\mathbf{q}\to 0} = -i\hat{\mathbf{k}}_\perp L, \tag{40}$$

$$\nabla_{\mathbf{q}}|\mathbf{e}_0 \cdot \hat{\mathbf{k}}_\perp^\pm|^2|_{\mathbf{q}\to 0} = \pm\frac{1}{k_\perp}\Re[e_0(e_0^* \cdot \mathbf{e}_\perp)], \tag{41}$$

$$\nabla_{\mathbf{q}}(\mathbf{e}_0 \cdot \hat{\mathbf{k}}_\perp^+)(e_0^* \cdot \hat{\mathbf{k}}_\perp^-)|_{\mathbf{q}\to 0} = \frac{i}{k_\perp}\Im[e_0(e_0^* \cdot \mathbf{e}_\perp)]. \tag{42}$$

Equations (41) and (42) involve a coupling between the beam polarization $\mathbf{e}_0$ and the momentum, which is eventually responsible for a spin Hall effect. This coupling effectively emerges as soon as the polarization of the incident beam is not purely TE or TM (the two principal axes of the problem), but is prepared in a superposition of the two. This can be explicitly seen by noting that for $\mathbf{e}_0 = \mathbf{e}_i^{\text{TE}}$, the scalar product $\mathbf{e}_0 \cdot \mathbf{k}_\perp \simeq \mathbf{e}_0 \cdot \mathbf{k}_0 \propto \mathbf{e}_x \cdot \mathbf{e}_y = 0$. On the other hand, if $\mathbf{e}_0 = \mathbf{e}_i^{\text{TM}}$ the gradients (41) and (42) provide a contribution that is aligned with the $x$ axis and is negligible compared to that of Eq. (40). For this reason, and to fix the ideas, from now on we consider for the incident beam polarization the simple mixture

$$\mathbf{e}_0 = \mathbf{e}_0^\sigma \equiv (\mathbf{e}_i^{\text{TM}} + i\sigma\mathbf{e}_i^{\text{TE}})/\sqrt{2}, \tag{43}$$

where $\sigma = \pm 1$ is the beam helicity. An initial state of this form describes a circularly polarized beam (left- or right-handed depending on the sign of $\sigma$). In this particular choice, where the ratio of coefficients in (43) is purely imaginary, $\Re[e_0(e_0^* \cdot \hat{k}_\perp)] = 0$ such that the SHE is entirely due to the term $\propto \Im[e_0(e_0^* \cdot \hat{k}_\perp)] \neq 0$. Note, however, that a superposition of TE and TM modes involving purely real coefficients (i.e. a linearly polarized incident beam) would also give rise to a SHE via the term $\propto \Re[e_0(e_0^* \cdot \hat{k}_\perp)] \neq 0$, a situation that was studied in [31]. For a polarization vector $e_0$ of the form (43), the term (42) yields a contribution to $\mathbf{R}_\perp$ that is aligned with the $y-$axis:

$$\mathbf{R}_\perp(L) = \hat{k}_0 L + R_y(L)e_y \,, \tag{44}$$

where

$$R_y(L) = -\frac{\sigma}{k_0}\left[1 - \frac{\cos L/2\ell_L}{\cosh L/2\ell_S}\right]. \tag{45}$$

The first term in the right-hand side of Eq. (44) describes the expected straight-line, ballistic evolution of the coherent beam along the direction of the transverse wave vector $\mathbf{k}_0$. This term is independent of the choice of the polarization vector $e_0$. The second term, on the other hand, corresponds to a helicity-dependent transverse shift of the order of $1/k_0$ in the $y$ direction. This shift, which was previously described in non-resonant transversally disordered media [30,31], constitutes a spin Hall effect of light in the random array. This spin Hall effect exhibits both a relaxation and harmonic oscillations, respectively controlled by the two longitudinal spatial scales

$$\ell_S \equiv [\Im(\Sigma_{TM} - \Sigma_{TE})/k]^{-1}, \quad \ell_L \equiv [\Re(\Sigma_{TM} - \Sigma_{TE})/k]^{-1}. \tag{46}$$

Note that unlike $\ell_{scat}$ which is by construction always positive, see Eq. (37), $\ell_S$ and $\ell_L$ can have an arbitrary sign due to their definition in terms of a difference of self-energies. Physically, the shift (45) stems from the interference between spin-orbit coupled photons propagating in the effective medium at the two spatial frequencies $k - \Sigma_{TE}/k$ and $k - \Sigma_{TM}/k$ [second term in Eq. (29)]. This interference involves both oscillation and relaxation effects because of the complex nature of the self energies $\Sigma_{TE,TM}$.

It should be noted that the general structure (45) of $R_y$ only relies on the peculiar uniaxial symmetry of the scattering medium [i.e. disordered in the plane $(x, y)$ but not along $z$]. In particular, the same expression holds in continuous, non-resonant dielectric random media [30,31], the microscopic details of the disorder only showing up through the parameters $\ell_S$ and $\ell_L$. What makes the discrete random array of special interest here is that, by exploiting the highly resonant character of the scatterers near $\omega_1$ for a TM excitation, it is possible to access a regime where $\ell_S$ and $\ell_L$ become *smaller* than the mean free path $\ell_{scat}$. This is in marked contrast with non-resonant media, where it was found that [30,31]

$$\ell_{S,L} \sim \ell_{scat}/\sin^2\theta, \tag{47}$$

so that the spin-Hall effect arises at a scale where the coherent mode has been exponentially attenuated by multiple scattering and is, consequently, not easily detectable.

To illustrate the above property, we show in Fig. 5(a) the ratios $\ell_{scat}/\ell_{S,L}$ as a function of detuning, in the close vicinity of the TM resonance. These ratios are computed by inserting Eqs. (25), (26) and (28) in the definitions (46). Observe that in the vicinity of this resonance, one can easily achieve $|\ell_{scat}/\ell_{S,L}| > 1$. The reason for this striking effect is the narrow character of the TM resonance at small angle of incidence (see Fig. 3), together with the fact that this resonance directly governs the magnitude of $\ell_{S,L}$ via $\Sigma_{TM}$ [see Eqs. (26) and (28)] but *not* of the mean free path $\ell_{scat}$ (which only depends on $\Sigma_{TE}$, i.e. on the lowest Mie resonance). Corresponding curves of the spin Hall shift $R_y$ vs. $L$ near the TM resonance are shown in Fig. 5(b). Exactly at resonance, the spin Hall effect appears at a scale smaller than the mean free path (whose position is indicated by the vertical dotted line). On the contrary, as one

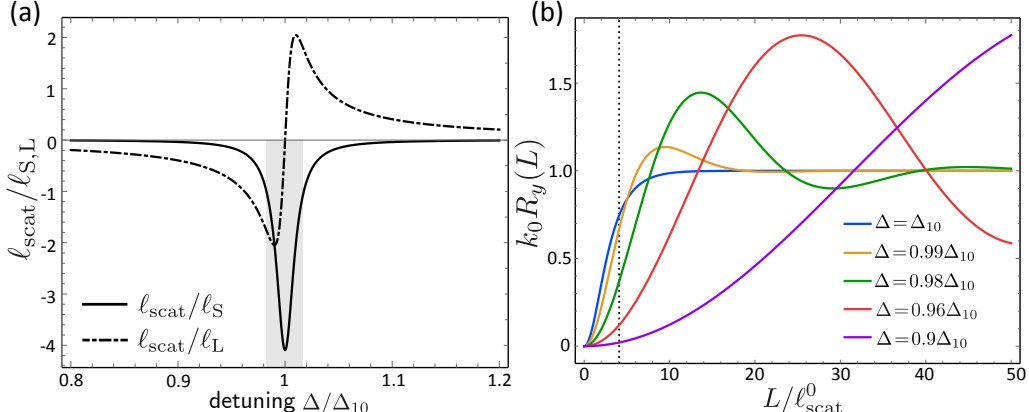

Figure 5: (a) Ratios $\ell_{\text{scat}}/\ell_{\text{S,L}}$ of the scattering mean free path along $z$ to the spin Hall mean free paths vs. frequency near the Mie resonance at $\omega = \omega_1$ [Here $\Delta = (\omega - \omega_0)/(\Gamma_0/2)$ is expressed in units of $\Delta_{10} = (\omega_1 - \omega_0)/(\Gamma_0/2)$; $\Delta/\Delta_{10} = 1$ thus corresponds to $\omega = \omega_1$]. Observe that in the close vicinity of the resonance there are frequency intervals where the ratio $|\ell_{\text{scat}}/\ell_{\text{S,L}}|$ exceeds unity. The shaded region, in particular, emphasizes the range of frequencies where $|\ell_{\text{scat}}/\ell_{\text{S}}| > 1$. (b) Transverse shift of the beam centroid as a function of $L$ for various detunings. The vertical dotted line indicates the location of the mean free path $\ell_{\text{scat}}$ (for the considered near-resonant detunings, $\ell_{\text{scat}}$ is nearly independent of $\Delta$). For all plots we set $m = 3$, $\theta = 0.05$ and the helicity $\sigma = -1$. The plot demonstrates that in the vicinity of the TM resonance the spin-Hall shift appears at a smaller scale than $\ell_{\text{scat}}$.

deviates from the resonance, $R_y$ becomes sizeable at a much larger scale than $\ell_{\text{scat}}$, with $\ell_{\text{S,L}}$ converging to the non-resonant result (47).

## 4.4 Asymptotic limits of the spin-orbit parameters

To gain a better insight on the behavior of the SHE described above, it is instructive to examine the analytical expressions of $\ell_{\text{S}}$ and $\ell_{\text{L}}$ in some characteristic limits. The explicit frequency dependence of these parameters is obtained by inserting Eqs. (25), (26) and (28) into the definitions (46):

$$\frac{1}{\ell_{\text{S}}} = \frac{4n}{k}\left[\sin^2\theta \frac{(\Gamma_0/2)^2(\omega/\omega_0)^4}{(\omega-\omega_0)^2 + (\Gamma_0/2)^2(\omega/\omega_0)^4} - \frac{(\Gamma_1/2)^2}{(\omega-\omega_1)^2 + (\Gamma_1/2)^2}\right], \tag{48}$$

$$\frac{1}{\ell_{\text{L}}} = \frac{4n}{k}\left[\frac{(\Gamma_1/2)(\omega-\omega_1)}{(\omega-\omega_1)^2 + (\Gamma_1/2)^2} - \sin^2\theta \frac{(\Gamma_0/2)(\omega-\omega_0)(\omega/\omega_0)^2}{(\omega-\omega_0)^2 + (\Gamma_0/2)^2(\omega/\omega_0)^4}\right]. \tag{49}$$

Let us first consider the static limit $\omega \to 0$, which formally corresponds to the model of non-resonant, continuous disordered media discussed in [30, 31]. Equation (48) gives

$$\ell_{\text{S}}(\omega \to 0) \simeq \frac{1}{\sin^2\theta}\frac{k}{4n}\left(\frac{2\omega_0}{\Gamma_0}\right)^2\left(\frac{\omega_0}{\omega}\right)^4 = \frac{\ell_{\text{scat}}(\omega \to 0)}{\sin^2\theta}, \tag{50}$$

where we have neglected the second term in the right-hand side of Eq. (48) using that $\Gamma_1 \sim \sin^2\theta\,\Gamma_0$, and we have used the expression (37) of the mean free path in the second equality. We thus recover the generic relation (47) for non-resonant materials.

Second, we examine the most interesting situation where the laser is tuned near the TM resonance, i.e. $\omega \simeq \omega_1$. In that case, the second term in the right-hand side of Eq. (48)

Table 1: Asymptotic values of the ratio $|\ell_{\text{scat}}/\ell_S|$ depending on the relative dispersion $\Delta\rho/\bar{\rho}$ of tube radii with respect to the quality factors $Q_0$ and $Q_1$ of the two lowest Mie resonances. When $\Delta\rho/\bar{\rho} \ll Q_1^{-1}$, the result (51) is recovered. The ratio decreases as soon as $\Delta\rho/\bar{\rho}$ exceeds $Q_1^{-1}$.

| | $\frac{\Delta\rho}{\bar{\rho}} \ll \frac{\Gamma_1}{2\omega_1} \equiv Q_1^{-1}$ | $Q_1^{-1} \ll \frac{\Delta\rho}{\bar{\rho}} \ll Q_0^{-1}$ | $Q_0^{-1} = \frac{\Gamma_0}{2\omega_0} \ll \frac{\Delta\rho}{\bar{\rho}}$ |
|---|---|---|---|
| $\left\|\frac{\ell_{\text{scat}}}{\ell_S}\right\|$ | $\frac{\Delta_{10}^2+(1+\Delta_{10}/Q_0)^4}{(1+\Delta_{10}/Q_0)^4}>1$ | $\frac{Q_1^{-1}}{\Delta\rho/\bar{\rho}}\frac{\Delta_{10}^2+(1+\Delta_{10}/Q_0)^4}{(1+\Delta_{10}/Q_0)^4}$ | $\frac{Q_0}{Q_1} \sim \sin^2\theta \ll 1$ |

becomes dominant and leads to

$$|\ell_S(\omega \simeq \omega_1)| \simeq \frac{k}{4n} = \ell_{\text{scat}}^0 < \ell_{\text{scat}} = \frac{k}{4n}\frac{\Delta_{10}^2 + (1+\Delta_{10}/Q_0)^4}{(1+\Delta_{10}/Q_0)^4}, \tag{51}$$

where $\Delta_{10} = (\omega_1 - \omega_0)/(\Gamma_0/2)$ is the (normalized) difference in resonance frequencies and $Q_0 = \omega_0/(\Gamma_0/2)$ is the quality factor of the lowest resonance. This confirms the result of the previous section, namely that the SHE occurs at a spatial scale shorter than the mean free path. Note, on the other hand, that $\ell_L \sim \ell_{\text{scat}}^0/\sin^2\theta \gg \ell_{\text{scat}}$ exactly at the TM resonance. This explains the absence of oscillations in the spin Hall shift visible in Fig. 5(b) for $\Delta = \Delta_{10}$.

In the case, finally, where the laser is tuned near the TE resonance ($\omega \simeq \omega_0$), we find from Eqs. (48) and (49) the same behavior as in the static limit: $\ell_{S,L}(\omega \simeq \omega_0) \simeq \ell_{\text{scat}}/\sin^2\theta \gg \ell_{\text{scat}}$, so that there is no significant SHE at the scale of the mean free path.

## 4.5 Impact of tube polydispersity

In the above analysis, we have assumed that all tubes forming the photonic array have identical radii. In practice, however, any realistic system will inevitably involve a finite dispersion of tube radii. Here we discuss the impact of this effect, expected to be the main limitation to the reduction of the spin Hall mean free path at the TM resonance. In the presence of a dispersion of tube radii, an inhomogeneous broadening of Mie resonances occurs. Near $\omega \simeq \omega_1$, the scattering and spin Hall mean free paths become

$$\frac{1}{\ell_{\text{scat}}} = -\int d\rho P(\rho)\frac{4n}{k}\text{Im}\frac{\Gamma_0/2(\omega_1/\omega_0)^2}{\omega - \omega_0 + i\Gamma_0/2(\omega_1/\omega_0)^2}, \tag{52}$$

and

$$\frac{1}{\ell_S} \simeq \int d\rho P(\rho)\frac{4n}{k}\text{Im}\frac{\Gamma_1/2}{\omega - \omega_1 + i\Gamma_1/2}, \tag{53}$$

where $P(\rho)$ is the distribution of tube radii $\rho$ and both the resonance frequencies and bandwidths implicitly depend on $\rho$ (see appendix B). These expressions suggest that the dispersion of radii will have a qualitative impact on the spin Hall effect as soon as the relative dispersion $\Delta\rho/\bar{\rho}$ becomes larger than the relative bandwidth of the TM resonance. As an example, we show in Table 1 the values of the ratio $|\ell_{\text{scat}}/\ell_S|$ computed for a Gaussian distribution function $P(\rho) = 1/(\sqrt{2\pi}\Delta\rho)\exp[-(\rho - \bar{\rho})^2/2\Delta\rho^2]$. When $\Delta\rho/\bar{\rho} \ll Q_1^{-1} \equiv \Gamma_1/(2\omega_1)$, we recover Eq. (51). In contrast, in the opposite regime $\Delta\rho/\bar{\rho} \gg Q_0^{-1} \equiv \Gamma_0/(2\omega_0)$ of large dispersion, $|\ell_{\text{scat}}/\ell_S| \sim \sin^2\theta$ becomes very small so that the benefit of the resonance is lost. At intermediate dispersions, finally, $|\ell_{\text{scat}}/\ell_S|$ is decreased but can remain close to one as long as $\Delta\rho/\bar{\rho} \ll Q_0^{-1}$.

## 4.6   Giant spin Hall effect

According to Eq. (44), the spin Hall shift in a random array has a maximum value $R_y \sim k_0^{-1}$. Although in the near paraxial regime $k_0/k \ll 1$ this scale greatly exceeds the optical wavelength, in practice it remains small compared to the beam width $w_0$ (recall that our approach assumes a collimated beam $k_0 w_0 \gg 1$). This makes the SHE still hardly visible, even by operating near the TM resonance. In order to achieve a *giant* SHE, the last step consists in combining the resonant excitation with a 'weak quantum measurement' [9, 11, 21, 22], where one performs a post-selection of the polarization of the transmitted light. The principle of this method in a random medium has been presented in [30, 31]. In short, it consists in starting from a linearly polarized beam $e_0 = e_i^{\text{TM}}$. Because this beam carries no helicity, it does not yield any SHE of the centroid (33) of the total field. The outcome, however, is different if one measures only a portion of the transmitted field using a post-selection polarizer $e_{\text{out}}$, so that

$$R_\perp(L) \equiv \frac{\int d\boldsymbol{r}_\perp \boldsymbol{r}_\perp |e_{\text{out}}^* \cdot \overline{\mathbf{E}}(\boldsymbol{r}_\perp, L)|^2}{\int d\boldsymbol{r}_\perp |e_{\text{out}}^* \cdot \overline{\mathbf{E}}(\boldsymbol{r}_\perp, L)|^2} \ . \tag{54}$$

In the weak-measurement scheme, an enhanced SHE is achieved by choosing a post-selection polarizer of the form $e_{\text{out}} \propto e_i^{\text{TE}} + i\delta e_i^{\text{TM}}$, where $|\delta| \ll 1$. This can be understood by decomposing the initial polarization state as $e_i^{\text{TM}} = 1/2(e_0^{\sigma=1} + e_0^{\sigma=-1})$ [where $e_0^\sigma$ is defined by Eq. (43)]. Upon crossing the random array, the two $\sigma = \pm 1$ field components experience spin Hall shifts of opposite signs, as illustrated in Fig. 6(b). As a consequence, the beam acquires a spatially-dependent polarization structure: the center of the beam is mostly linearly polarized (with polarization $e_i^{\text{TM}}$), whereas its far tails are circularly polarized with opposite helicities. Filtering the transmitted field with the polarizer $e_{\text{out}}$, has then two consequences: (i) the real part of $e_{\text{out}}$, perpendicular to $e_i^{\text{TM}}$, eliminates the non-shifted central part of the beam and (ii) the

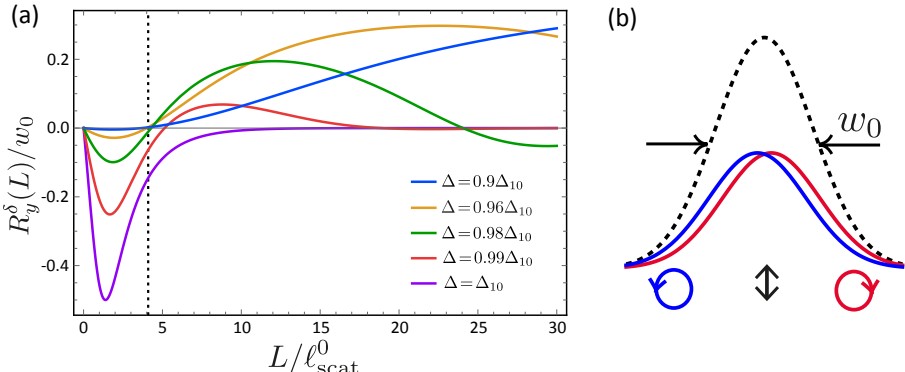

Figure 6: (a) Weak-measurement-based spin Hall shift $R_y^\delta(L)$ [Eq. (55)] vs. $L$ for various detunings, in the vicinity of the TM resonance. The vertical dotted line indicates the position of the mean free path $\ell_{\text{scat}}$. For all plots we set $m = 3$, $\theta = 0.05$ and $\delta = -(k_0 w_0)^{-1}$. As in Fig. 5, the shift appears at the same scale of $\ell_{\text{scat}}$, but its maximum value is now of the order of the beam width $w_0$ thanks to the post-selection scheme. (b) Principle of the weak measurement. When starting from a linearly polarized state, its $\sigma = \pm 1$ components are shifted in opposite directions, and the beam acquires a spatially-dependent polarization structure, its center being linearly polarized and its far tails being circularly polarized with opposite helicities. The weak measurement consists in selecting out one of the two tails using a conveniently chosen post-selection polarizer.

imaginary part of $e_{\text{out}}$ selects out one of the two tails and eliminates the other, depending on the sign of $\delta$. A calculation similar to that of Sec. 4 precisely yields for the component $R_y^\delta(L)$ of $\mathbf{R}_\perp$ along the $y-$axis:

$$R_y^\delta(L) = -\frac{\delta}{k_0} \frac{1 - e^{-L/2\ell_S}\cos(L/2\ell_L)}{\delta^2 + \left(\frac{1}{k_0 w_0}\right)^2 |1 - e^{-L/2\ell_S}e^{iL/2\ell_L}|^2} \ . \tag{55}$$

As compared to Eq. (45), which does not involve any post-selection, the shift (55) is now of the order of the beam width $w_0$ if one chooses $|\delta| = 1/(k_0 w_0)$. Equation (55) is displayed in Fig. 6 for various detunings in the close vicinity of the TM Mie resonance. Again, the spin Hall shift occurs at a smaller scale than the mean free path (indicated by a vertical dotted line), but contrary to Fig. 5(b) its maximum value is now of the order of $w_0$. This result, which constitutes the main finding of the paper, can be seen as a *giant* SHE in a random array, where the spin Hall shift is both macroscopic and occurs at a $z-$scale where the coherent mode is visible.

## 5  Spin-orbit coupled flash of light

In the previous section, we have shown that a giant SHE can emerge in the coherent mode by operating in the close vicinity of the TM resonance. This manifests itself by a spatial shift of the beam in the transverse plane of the random array, visible for optically-thin arrays of thickness $L \sim \ell_{\text{scat}}$ of the order of the scattering mean free path. The next natural question is whether the SHE can also be observed in an optically-*thick* random array of length $L \gg \ell_{\text{scat}}$. In the case of a stationary laser excitation considered so far, this is clearly hopeless due to the exponential attenuation of the coherent mode as $\exp(-L/\ell_{\text{scat}})$. For a *time-dependent* excitation, on the other hand, the resonant character of the system makes this possible by exploiting the 'coherent flash' phenomenon, as we now explain.

We again consider an incident laser beam of the form (22), but now suppose that the beam is suddenly switched off at some time $t = 0$:

$$\mathbf{E}(\mathbf{k}_\perp, z=0, t) = \sqrt{2\pi}w_0 \exp[-(\mathbf{k}_\perp - \mathbf{k}_0)^2 w_0^2/4 - i\omega_l]e_0\theta(-t), \tag{56}$$

where $\theta(t)$ refers to the Heaviside theta function and $\omega_l$ is the laser carrier frequency. In resonant, optically-thick media, such an abrupt extinction of the laser excitation is known to induce a coherent flash of light of the coherent mode intensity at $t = 0^+$ [42–44]. The origin of this effect can be understood from the decomposition (15) of the total field

$$\mathbf{E} = \mathbf{E}_{\text{in}} + \mathbf{E}_s, \tag{57}$$

as a sum of the incoming and scattered fields, $\mathbf{E}_{\text{in}}$ and $\mathbf{E}_s$, respectively. The coherently transmitted field stems from the destructive interference between $\mathbf{E}_{\text{in}}$ and the part of $\mathbf{E}_s$ scattered in the forward direction. If the laser is switched off, this interference is suppressed while the resonant scatterers continue to radiate over a time scale of the order of the inverse of the resonance width, leading to a flash of light. This can be mathematically formulated as follows. When the laser is switched off, $\mathbf{E}_{\text{in}}(t = 0^+) = 0$ such that $\mathbf{E}(t = 0^+) = \mathbf{E}_s(t = 0^+)$. On the other hand, if the medium is optically thick, $\mathbf{E}(t = 0^-) \simeq 0$ is exponentially attenuated, such that $\mathbf{E}_s(t = 0^-) = -\mathbf{E}_{\text{in}}(t = 0^-)$. Combining these relations assuming the continuity of the scattered field at $t = 0$ leads to $\mathbf{E}(t = 0^+) = -\mathbf{E}_{\text{in}}$, i.e. a flash of light of same intensity as the incoming beam.

Coherent flashes of light in optically-thick media have been theoretically described and experimentally observed in resonant cold atomic gases [42]. Later on, it was also shown that

flashes of intensity *larger* than that of the incoming laser could even be achieved by operating slightly away from resonance, with a maximum theoretical value of $4|\mathbf{E}_{\text{in}}|^2$ stemming from the general inequality $|\mathbf{E}_{\text{in}} + \mathbf{E}_s|^2 \leq |\mathbf{E}_{\text{in}}|^2$ imposed by energy conservation [43].

The coherent flash is expected to take place in our system as well, due to the resonant nature of the scatterers. This offers the possibility to observe a time-dependent SHE in optically-thick random arrays, provided the associated spin Hall shift exists on the time scale where the flash occurs. To describe this problem, we insert the relation

$$\exp(-i\omega_l t)\theta(-t) = \int \frac{d\omega}{2\pi} e^{-i\omega t}\left[ \pi\delta(\omega - \omega_l) - ip.v.\left(\frac{1}{\omega - \omega_l}\right)\right] \tag{58}$$

into Eq. (56), and make use of the temporal version of Eq. (29), which reads:

$$\overline{\mathbf{E}}(\mathbf{k}_\perp, z = L, t) = \sqrt{2\pi w_0^2}\exp\left[ -\frac{w_0^2}{4}(\mathbf{k}_\perp - \mathbf{k}_0)^2\right]\int \frac{d\omega}{2\pi} e^{-i\omega t}\left[ \pi\delta(\omega - \omega_l) - ip.v.\left(\frac{1}{\omega - \omega_l}\right)\right]$$
$$\times \left[ e^{-i\Sigma_{\text{TE}}L/2k_z}\mathbf{e}_0 + \left( e^{-i\Sigma_{\text{TM}}L/2k_z} - e^{-i\Sigma_{\text{TE}}L/2k_z}\right)\mathbf{p}(\mathbf{k}_\perp)\right]. \tag{59}$$

The space-time intensity distribution of the coherent mode follows from

$$I(\mathbf{r}_\perp, L, t) = \int \frac{d^2\mathbf{k}_\perp}{(2\pi)^2}\int \frac{d^2\mathbf{q}}{(2\pi)^2} e^{i\mathbf{q}\cdot\mathbf{r}_\perp}\overline{\mathbf{E}}(\mathbf{k}_\perp^+, L, t)\cdot\overline{\mathbf{E}}^*(\mathbf{k}_\perp^-, L, t), \tag{60}$$

with $\mathbf{k}_\perp^\pm = \mathbf{k}_\perp \pm \mathbf{q}/2$. Following the same lines as in Sec. 4, we end up with

$$I(\mathbf{r}_\perp, L, t) \simeq I(L, t)e^{-2|\mathbf{r}_\perp - \mathbf{R}_\perp(L,t)|^2/w_0^2}, \tag{61}$$

where

$$I(L, t) \equiv I_0\int \frac{d^2\mathbf{k}_\perp}{(2\pi)^2}|\overline{\mathbf{E}}(\mathbf{k}_\perp, L, t)|^2 = I(\mathbf{R}_\perp(L), L, t), \tag{62}$$

with $I_0 = 2/(\pi w_0^2)$, and

$$\mathbf{R}_\perp(L, t) = \frac{i\int \frac{d^2\mathbf{K}}{(2\pi)^2}\nabla_\mathbf{q}[\overline{\mathbf{E}}(\mathbf{K}_+, L, t)\cdot\overline{\mathbf{E}}^*(\mathbf{K}_-, L, t)]_{\mathbf{q}\to 0}}{\int \frac{d^2\mathbf{K}}{(2\pi)^2}|\overline{\mathbf{E}}(\mathbf{K}, L, t)|^2} \equiv \frac{\int d^2\mathbf{r}_\perp \mathbf{r}_\perp I(\mathbf{r}_\perp, L, t)}{\int d\mathbf{r}_\perp I(\mathbf{r}_\perp, L, t)}. \tag{63}$$

Equations (61, 62, 63) are the time-dependent versions of Eqs. (33, 34, 35). To evaluate the coherent mode intensity (62) and the beam centroid (63), we need to perform the Fourier transforms with respect to $\omega$ coming from Eq. (59), taking into account the explicit frequency dependence of $\Sigma_{\text{TE,TM}}$. To this aim and for the sake of simplicity, we drop the quadratic frequency corrections in Eq. (25) and use:

$$\Sigma_{\text{TE}}(\omega) = \frac{4n\Gamma_0/2}{\omega - \omega_0 + i\Gamma_0/2},$$
$$\Sigma_{\text{TM}}(\omega) = \frac{4n\Gamma_1/2}{\omega - \omega_0 + i\Gamma_1/2} + \frac{4n\Gamma_0/2\cos^2\theta}{\omega - \omega_0 + i\Gamma_0/2}. \tag{64}$$

This allows us to derive closed formulas for $I(L, t)$ and $\mathbf{R}_\perp(L, t)$, which can be used for the numerical simulations. These expressions are a little cumbersome and are given in Appendix C for clarity.

We first show in Fig. 7(a) the coherent mode intensity vs. time following the laser extinction for three values of the detuning $\Delta = (\omega_l - \omega_0)/(\Gamma_0/2)$, setting the optical thickness $L/\ell_{\text{scat}}^0$

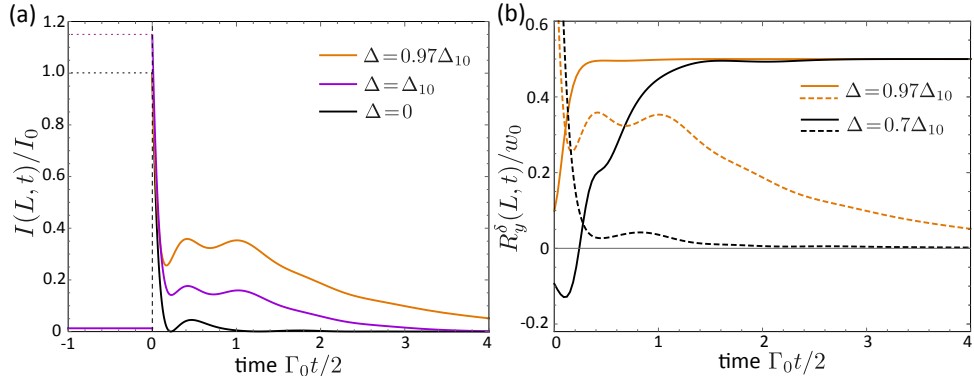

Figure 7: (a) Time evolution of the coherent mode intensity $I(L,t)$ transmitted through the random array, for three values of the detuning $\Delta = (\omega_l - \omega_0)/(\Gamma_0/2)$. In each case the incident laser is switched off at $t = 0$, which results in a flash of light at $t = 0^+$. In the vicinity of the TM resonance ($\Delta = \Delta_{10}$), the flash exhibits a long-lived tail. (b) Time evolution of the macroscopic spin Hall shift, based on the weak-measurement scheme with $\delta = -(k_0 w_0)^{-1}$, for two values of the detuning (solid curves). The corresponding intensities $I(L,t)/I_0$ are superimposed (dashed curves, same vertical scale). Near the TM resonance, $\Delta \simeq \Delta_{10}$, a large spin Hall shift appears over a time window where the coherent flash is still significant. For all plots we set $m = 3$, $\theta = 0.05$ and an optical thickness $L/\ell_{\text{scat}}^0 = 15$.

to a relatively large value. At $\Delta = 0$, we observe a coherent flash of maximum amplitude $I = I_0$ decaying over a typical time scale $\sim \Gamma_0^{-1} \ell_{\text{scat}}/L$, i.e. a fraction of the lifetime of the TE Mie resonance. This behavior is characteristic of the cooperative nature of the flash, pointed out in [42, 43] in the context of resonant cold atomic gases: when the medium is optically thick, the deepest scatterers are not only excited by the incoming laser but by the superposition of the incoming laser and the field radiated by the scatterers in shallower layers. This results in an effective loss of coherence corresponding to an increased linewidth.

On the other hand, at detunings $\Delta \simeq \Delta_{10}$ the temporal shape of the coherent intensity acquires a bimodal structure, with a short transient flash followed by a slowly decaying tail that persists up to the much longer time scale $\sim \Gamma_1^{-1} \ell_{\text{scat}}/L \sim \Gamma_0^{-1} \ell_{\text{scat}}/(L \sin^2 \theta)$. This long-lived tail is a direct manifestation of the resonance at $\omega_1$, whose contribution to $I(L,t)$ becomes significant when $\omega_l \sim \omega_1$ and adds up to the contribution of the resonance at $\omega_0$. In this doubly-resonant regime, the magnitude of the flash also slightly exceeds $I_0$.

Following the laser extinction, not only the intensity but also the spin Hall shift undergoes a temporal evolution. Its analytical expression is given in Appendix C, in both cases where a polarization post-selection is performed or not. We show in Fig. 7(b) its macroscopic value $R_y^\delta(L,t)$ along the $y-$axis obtained using the weak-measurement procedure explained in Sec. 4.6, for two different detunings and a fixed value $L/\ell_{\text{scat}}^0 \gg 1$ of the optical thickness. Near the TM resonance ($\Delta \simeq \Delta_{10}$), the shift reaches a value $w_0/2$ over a time scale $\Gamma_0^{-1} \ell_{\text{scat}}/L$, much shorter than the duration $\Gamma_1^{-1} \ell_{\text{scat}}/L$ of the coherent flash. This demonstrates the relevance of exploiting coherent flashes in resonant media to observe a sizeable value of the SHE at large optical thicknesses. In contrast, Fig. 7(b) shows that away from the TM resonance the shift takes a longer and longer time to appear while the flash duration becomes shorter and shorter.

# 6 Conclusion and outlook

In this paper, we have described the propagation of light in transversally disordered photonic arrays, and have shown evidence for an enhanced spin Hall effect of light in the vicinity of the second Mie resonance of the system. This resonance is associated with the TM polarization component of the beam impinging on the medium at an angle of incidence $\theta$, and exhibits an ultra-narrow spectral width scaling as $\theta^2$. This geometrical property gives rise to a SHE mean free path $\ell_{\rm S}$ smaller than the scattering mean free path $\ell_{\rm scat}$, in strong contrast with non-resonant materials for which $\ell_{\rm S} \sim \ell_{\rm scat}/\theta^2 \gg \ell_{\rm scat}$. This implies that the SHE occurs at a spatial scale where the coherent mode is still significant. Furthermore, we have provided a temporal description of the SHE following the abrupt switch off of the incoming beam. In this scenario, the resonant nature of the photonic array leads to a flash of light in the coherently transmitted signal, which allows to detect the coherent mode in the regime of large optical thickness. Again in the vicinity of the TM resonance, we have shown evidence for a long-lived flash of light together with the emergence of a sizeable spin Hall effect in the same time window.

In practice, the proposed scheme should bring the SHE of light in a random medium within reach of experimental detection. To this aim, a good candidate could be random arrays imprinted on glass with femto-second writing beams [48, 49]. In those systems, the refractive index ratio $m$ is typically close to 1. For a monodisperse array of tubes, this implies a ratio $\ell_{\rm S}/\ell_{\rm scat} \sim 1$ near the TM resonance, so that the resonant SHE is still present. The only requirement would then to make the mean free path not too large compared to the system size. If this requirement is fulfilled, the main remaining limitation might be the dispersion of tube radii, which as we have shown starts to increase $\ell_{\rm S}$ when the relative dispersion of radii exceeds the inverse of the quality factor of the TM resonance.

The SHE described in the present work only arises due to the statistical uniaxial anisotropy of the transverse disorder. Presumably, however, other types of uniaxial anisotropy, such as a three-dimensional correlated disorder with anisotropic (uniaxial) correlation function might also support spin-orbit phenomena and thus constitute natural extensions of this work. As opposed to photons in the coherent mode that propagate in the forward direction, it would be also important to clarify the role of spin-orbit corrections for the photons scattered in other directions by the disorder, known to generically dominate the multiple scattering signal at large optical thickness. Other interesting directions of research would be to provide a geometric-phase description of spin-orbit interactions in a random medium, following similar approaches proposed in deterministic systems [14, 50, 51], or to explore optical analogues of the quantum spin Hall effect in the presence of disorder [52, 53].

# Acknowledgements

This project has received financial support from the CNRS through the 80'Prime program, and from the Agence Nationale de la Recherche (grant ANR-19-CE30-0028-01 CONFOCAL).

# A Scattering by a cylinder: Mie coefficients

Mie theory provides an exact solution to electromagnetic-wave scattering by objects of arbitrary size. In the case of an infinitely-long cylinder, analytical expressions for the Mie coeffi-

cients entering Eq. (7) are available [45]. We report them here:

$$a_{nI} = \frac{C_n V_n - B_n D_n}{W_n V_n + i D_n^2}, \quad b_{nI} = \frac{W_n B_n + i D_n C_n}{W_n V_n + i D_n^2}, \quad a_{nII} = -\frac{A_n V_n - i C_n D_n}{W_n V_n + i D_n^2}, \quad \text{(A.1)}$$

where

$$\begin{aligned}
A_n &= i\xi\big[\xi J_n'(\eta)J_n(\xi) - \eta J_n(\eta)J_n'(\xi)\big], \\
B_n &= \xi\big[m^2 \xi J_n'(\eta)J_n(\xi) - \eta J_n(\eta)J_n'(\xi)\big], \\
C_n &= n\cos\theta\, \eta J_n(\eta)J_n(\xi)(\xi^2/\eta^2 - 1), \\
D_n &= n\cos\theta\, \eta J_n(\eta)H_n^{(1)}(\xi)(\xi^2/\eta^2 - 1), \\
V_n &= \xi\big[m^2 \xi J_n'(\eta)H_n^{(1)}(\xi) - \eta J_n(\eta)H_n^{(1)\prime}(\xi)\big], \\
W_n &= i\xi\big[\eta J_n(\eta)H_n^{(1)\prime}(\xi) - \xi J_n'(\eta)H_n^{(1)}(\xi)\big],
\end{aligned} \quad \text{(A.2)}$$

with $\xi = k\rho\sin\theta$, $\eta = k\rho\sqrt{m^2 - \cos^2\theta}$, and $J_n$ and $H_n^{(1)}$ are the Bessel and Hankel functions of the first kind, respectively.

# B   Parameters of lowest Mie resonances

In this appendix, we provide the analytical expressions of the resonance frequencies $\omega_{0,1}$ and widths $\Gamma_{0,1}$ of the two lowest Mie resonances of a dielectric cylinder. These expressions are obtained by linearizing the Bessel functions in Eq. (A.2) in the vicinity of the first zeros of the denominators of $b_{0I}$ and $a_{1I}$. For the lowest Mie resonance we find

$$\omega_0 = \frac{c_0/8\rho}{\gamma + \ln\frac{a\sin\theta}{2\sqrt{m^2-1}}}\left[\frac{adm^2}{c\sqrt{m^2-1}} - \sqrt{\frac{a^2 d^2 m^4}{c^2(m^2-1)} - \left(8 + \frac{4a^2 dm^2}{c(m^2-1)}\right)\left(4\gamma + 4\ln\frac{a\sin\theta}{2\sqrt{m^2-1}}\right)}\right], \quad \text{(B.1)}$$

and

$$\Gamma_0 = \frac{\rho}{c_0}\frac{4\pi\omega_0^2}{\sqrt{\frac{a^2 d^2 m^4}{c^2(m^2-1)} - \left(8 + \frac{4a^2 dm^2}{c(m^2-1)}\right)\left(4\gamma + 4\ln\frac{a\sin\theta}{2\sqrt{m^2-1}}\right)}}, \quad \text{(B.2)}$$

where $\gamma$ is the Euler's constant and $a = 1.84118378$, $b = 0.2051107$, $c = 0.581865$ and $d = 0.8204428$. We also recall that $\theta$ is the angle of incidence of the laser on the cylinder, $\rho$ is the cylinder radius and $m$ is the refractive index of the cylinder with respect to the surrounding medium. Similarly, for the TM Mie resonance we find

$$\omega_1 = \frac{c_0}{\rho}\left[\frac{\alpha}{\sqrt{m^2-1}} + \frac{m\alpha\sin^2\theta}{m^2-1}\left(\gamma + \ln\frac{\alpha\sin\theta}{2\sqrt{m^2-1}}\right)\right], \quad \text{(B.3)}$$

and

$$\Gamma_1 = \frac{c_0}{\rho}\frac{m\alpha\pi\sin^2\theta}{m^2-1}, \quad \text{(B.4)}$$

where $\alpha = 2.4048$. In particular, at very small angle Eq. (B.3) reduces to Eq. (11) of the main text. In the limit $\theta \to 0$ the width $\Gamma_0$ goes to zero very slowly, whereas $\Gamma_1$ vanishes very fast due to the $\sin^2\theta$ scaling.

## C  Coherent mode intensity and spin Hall shift following laser extinction

To compute the coherent-mode intensity $I(L, t)$ following a laser extinction, we insert Eq. (59) into Eq. (62) and perform the integral over momentum. This gives

$$
I(L, t) = I_0 \left| \int \frac{d\omega}{2\pi} e^{-i\omega t} \left[ \pi \delta(\omega - \omega_l) - i p.v. \left( \frac{1}{\omega - \omega_l} \right) \right] \right.
$$
$$
\left. \times \left[ e^{-i\Sigma_{\mathrm{TE}} L/2k_z} \boldsymbol{e}_0 - \left( e^{-i\Sigma_{\mathrm{TE}} L/2k_z} - e^{-i\Sigma_{\mathrm{TM}} L/2k_z} \right) \boldsymbol{p}(\boldsymbol{k}_0) \right] \right|^2 . \tag{C.1}
$$

Then we express the exponential terms $\exp(-i\Sigma_{\mathrm{TE,TM}} L/2k_z)$ as infinite series using Eqs. (64), for instance:

$$
\exp(-i\Sigma_{\mathrm{TE}} L/2k_z) = \sum_{p=0}^{\infty} \frac{1}{p!} \left( -\frac{iz}{2\ell_{\mathrm{scat}}^0} \frac{\Gamma_0/2}{\omega - \omega_0 + i\Gamma_0/2} \right)^p , \tag{C.2}
$$

and perform the frequency integrals in Eq. (C.1) using the residue theorem. The calculation is tedious but straightforward. It gives

$$
I(L, t) = \frac{I_0}{2} \left[ |F(L, t)|^2 + |G(L, t)|^2 \right] , \tag{C.3}
$$

where

$$
G(L, t) = \sum_{p=0}^{\infty} \frac{1}{p!} \left( -\frac{L}{2\ell_{\mathrm{scat}}^0} \frac{i\Gamma_0/2}{\delta_0 + i\Gamma_0/2} \right)^p \frac{\Gamma(p, (\Gamma_0/2 - i\delta_0)t)}{\Gamma(p)} , \tag{C.4}
$$

and

$$
\begin{aligned}
F(L, t) = &\exp\left( -\frac{L}{2\ell_{\mathrm{scat}}^0} \frac{i\Gamma_1/2}{-\Delta\omega - i\Delta\Gamma/2} \right) \sum_{p=0}^{\infty} \frac{1}{p!} \left( -\frac{L\cos^2\theta}{2\ell_{\mathrm{scat}}^0} \frac{i\Gamma_0/2}{\delta_0 + i\Gamma_0/2} \right)^p \frac{\Gamma(p, (\Gamma_0/2 - i\delta_0)t)}{\Gamma(p)} \\
&+ \exp\left( -\frac{L\cos^2\theta}{2\ell_{\mathrm{scat}}^0} \frac{i\Gamma_0/2}{\Delta\omega + i\Delta\Gamma/2} \right) \sum_{p=0}^{\infty} \frac{1}{p!} \left( -\frac{L}{2\ell_{\mathrm{scat}}^0} \frac{i\Gamma_1/2}{\delta_1 + i\Gamma_1/2} \right)^p \frac{\Gamma(p, (\Gamma_1/2 - i\delta_1)t)}{\Gamma(p)} \\
&+ \sum_{p=0}^{\infty} \sum_{s=1}^{p-1} \frac{1}{p!} \left( -\frac{L\cos^2\theta}{2\ell_{\mathrm{scat}}^0} \frac{i\Gamma_0/2}{\delta_0 + i\Gamma_0/2} \right)^p \frac{\Gamma(p - s, (\Gamma_0/2 - i\delta_0)t)}{\Gamma(p - s)} \\
&\quad \times \left( \frac{\delta_0 + i\Gamma_0/2}{\Delta\omega + i\Delta\Gamma/2} \right)^s M_s\left( -\frac{L}{2\ell_{\mathrm{scat}}^0} \frac{i\Gamma_1/2}{-\Delta\omega - i\Delta\Gamma/2} \right) \\
&+ \sum_{p=0}^{\infty} \sum_{s=1}^{p-1} \frac{1}{p!} \left( -\frac{L}{2z_s^0} \frac{i\Gamma_1/2}{\delta_1 + i\Gamma_1/2} \right)^p \frac{\Gamma(p - s, (\Gamma_1/2 - i\delta_1)t)}{\Gamma(p - s)} \\
&\quad \times \left( \frac{\delta_1 + i\Gamma_1/2}{-\Delta\omega - i\Delta\Gamma/2} \right)^s M_s\left( -\frac{L\cos^2\theta}{2\ell_{\mathrm{scat}}^0} \frac{i\Gamma_0/2}{\Delta\omega + i\Delta\Gamma/2} \right) .
\end{aligned} \tag{C.5}
$$

Here $\Gamma(n, x)$ is the upper incomplete Gamma function, $M_s(x) = {}_1F_1(1 + s, 2, x)x$ with ${}_1F_1$ being the Kummer's confluent hypergeometric function, and the various detunings are defined as

$$
\begin{aligned}
\delta_0 &= \omega_l - \omega_0 , \\
\delta_1 &= \omega_l - \omega_1 , \\
\Delta\omega &= \omega_1 - \omega_0 , \\
\Delta\Gamma &= \Gamma_1 - \Gamma_0 .
\end{aligned} \tag{C.6}
$$

The calculation of the spin Hall shift of the coherent mode after the laser extinction is performed using Eq. (63) and follows the same strategy as in the stationary case. We find:

$$R_y(L,t) = -\frac{\sigma}{k_0}\left[1 - \frac{2\Re[F^*(L,t)G(L,t)]}{|F(L,t)|^2 + |G(L,t)|^2}\right]. \tag{C.7}$$

The time-dependent spin Hall shift following from the weak-measurement procedure with $\boldsymbol{e}_{\text{out}} \propto \boldsymbol{e}_i^{\text{TE}} + i\delta \boldsymbol{e}_i^{\text{TM}}$ as a post-selection polarizer, finally, is given by:

$$R_y^\delta(L,t) = -\frac{\delta}{k_0} \frac{|F(L,t)|^2 - \Re[F^*(L,t)G(L,t)]}{\delta^2|F(L,t)|^2 + \left(\frac{1}{k_0 w_0}\right)^2|G(L,t) - F(L,t)|^2}. \tag{C.8}$$

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
