# Peer review of "Resonant Spin Hall Effect of Light in Random Photonic Arrays"

_SciPost Physics, doi:SciPost Phys. 14, 104 (2023)_

## Round 1 · Referee Report · Anonymous (Referee 1) · 2022-12-12

Strengths

  1. Is based on a very basic electromagnetic configuration

  2. The proposed system offers a significant reinforcement of the spin Hall signal, an effect of interest both in optics and in condensed matter physics.

  3. The presentation is reasonably in spite of the highly technical nature of the manuscript

Weaknesses

  1. An explicit discussion of the experimental feasibility of the proposal should be included, along with a critical summary of possible experimental difficulties.

  2. Referencing to previous literature and putting the results in a wider context needs improving

  3. The topic and the presentation are quite technical

Report

The manuscript by Carlini and Cherroret reports a theoretical study of light propagation through a resonant disordered medium. In particular, it points out siginificant advantages of such a system in view of observing an optical version of the spin-Hall effect. Even though the manuscript addresses a quite specialized topic, it is reasonably accessible also to non-specialists like me. The basic mechanism of te proposal is based on a remarkably simple yet subtle and not so well-known feature of electromagnetism such as the Mie resonances, and exploits their features to get a dramatic enhancement of the optical SHE.
On this basis, and given also the possible connections with hot topics in condensed matter physics, I may anticipate that this work has the potential to attract the interest of a quite diverse audience. As such, it may eventually deserve publication on SciPost.

Before that, however, the authors should take into due consideration the following remarks on both the presentation and the content of the manuscript.

-A non-expert reader may not appreciate the importance of working with circularly polarized light to observe the SHE. For instance, the sentence "For a reason that...populating both modes." on pag.10-11 is misleading as it gives the impression that any superposition of TE and TM makes the job independently of the real/complex value of the coefficients. The sentence should be clarified and extended, and it may be also useful to spend a couple of sentence on the symmetries that need being broken in order to observe the SHE.

-I feel slightly unconfortable in directly transposing the Green function of an isolated cylinder to an ensemble of many cylinders, in particular for what concerns the linewidth of the Mie resonances. In the case of an ordered array, the linewidth in fact disappears and polariton modes with infinite lifetime are formed upon mixing the light and matter resonances. I guess that this does not happen in a disordered medium where scattering processes can instead occur and wavevector is no longer a good quantum number.
I may anticipate that naive readers like me may appreciate some discussion along these lines.

-before eq.(35): the authors choose as a benchmark for all length scales along z the TE scattering mean free path \ell_{scat}, calculated including the 0 resonance only and taken at the operating frequency \omega. Since the SHE occurs in the vicinity of the 1 resonance, I would have found more natural to use some other length scale that specifically includes also the 1 resonance. Or there is some general argument to justify that \ell_{scat} (or better \ell_{scat}^0) is a valid choice in all cases ?

-along similar lines, I find the authors' choice for x-axis of fig.4 a bit unfortunate, in particular for what concerns the different locations of the narrow peak on the different curves. From what the authors say in the text, I understand that this is not due to a real motion of the resonance, but to the angle-dependence of the reference point omega_0. I suggest the authors to adopt in the whole manuscript a more transparent notation to indicate the frequencies.

-I suspect some problem with the y-axis label of fig.5(a). I would have expected that all \ell's be positive quantities. The caption also contains a suspicious statement "the ratios exceed unity in the vicinity of the resonance" which I can not visualize on the curves.

-eq.(21): I don't find the definition of \bar{G}(k_\perp,z=L). The authors should add it in explicit form somewhere.

-on pag.17, it could be worth adding some physical interpretation to the formulas for the time scales of the coherent flash, \Gamma_0^{-1} \ell_{scat}/L and \Gamma_1^{-1} \ell_{scat}/L. In particular, the inverse proportionality to the sample length L is very counterintuitive and deserves justification.

-in the conclusions, I do not understand what the authors mean by "account for the spin-orbit corrections in the multiple-scattering signal". I thought that SO-coupling was already included in the present work. Or am I missing something important?

-The authors should complete the introduction and the bibliography with some discussion of earlier works on the use of Mie resonances to realize photonic crystals and/or metamaterials with exotic properties. I have heard about such developments at several conferences and I am sure there is abundant literature on the topic. These revisions are essential to put the work in a proper context.

-As far as I understand, the authors are assuming that the cylinders are located at random positions but have identical radii. I expect that any spread in the cylinder radius will result in a inhomogeneous broadening of the narrow Mie resonance, a quick decoherence of the flash of light signal, and, finally, to a significant reduction of the SHE magnitude. Most likely, this is the most serious experimental difficulty of the authors' proposal. Even though this is a theoretical proposal,
for the sake of completeness it is important that the authors provide some estimate of the robustness of their predictions against a finite spread in the cylinder radii and the discussion of the actual experimental feasibility.

-On pag.7, the authors present in a quite rapid way the assumptions they are doing on the statistics of the cylinder positions. What does "uniformly distributed mean" ? How is this condition compatible with the fact that two cylinders can not overlap? What would be the next corrections to this simplified calculations?

-How necessary is to have a disordered medium to observed the SHE. Would an ordered lattice of cylinders be sufficient to observe a similar or disorder is essential? I think that non-expert readers would appreciate having a bit of discussion in this sense in the introduction.

Requested changes

  1. Revise the presentation to clarify some technical points

  2. Discuss the limitations of their proposal and the most serious experimental difficulties.

  3. Put the manuscript in the context of the existing literature providing a fair account of previous works.

  4. Try to highlight the general interest of the optical spin Hall effect in a wider context.

  • validity: high
  • significance: high
  • originality: high
  • clarity: high
  • formatting: excellent
  • grammar: perfect

Author:  Nicolas Cherroret  on 2023-01-16  [id 3240]

(in reply to Report 1 on 2022-12-12)
Category:
answer to question

We thank the referee for their positive appreciation of our manuscript. Below with carefully address their questions and comments.

1) A non-expert reader may not appreciate the importance of working with circularly polarized light to observe the SHE. For instance, the sentence "For a reason that...populating both modes." on pag.10-11 is misleading as it gives the impression that any superposition of TE and TM makes the job independently of the real/complex value of the coefficients. The sentence should be clarified and extended, and it may be also useful to spend a couple of sentence on the symmetries that need being broken in order to observe the SHE.

Response: In fact, in our system the existence of a spin Hall effect does not require the use of circularly (or even elliptically) polarized light. For instance, for a purely real superposition of the TE and TM modes in Eq. (43) (which could correspond to a linearly polarized incident beam), a lateral shift also appears because of the term (42), even though the SHE has slightly different properties in that case. Therefore, strictly speaking the only condition for observing a SHE is that the initial polarization does not coincide with the two principal axes of the problem, $e_\text{TE}$ and $e_\text{TM}$. However, we fully agree with the referee that this point was not clearly explained in the manuscript. In the new version, we have revisited and extended the paragraphs around Eq. (43) to discuss this better, following the referee’s suggestion. We have also modified the end of the second paragraph of the introduction to clarify the role of the polarization of the incident beam. The general question of which type of symmetry needs to be broken in a disordered system to guarantee the existence of a SHE is not completely clear to us and probably goes beyond the scope of our work. Nevertheless, according to our study it seems that a disorder with (statistical) uniaxial anisotropy is a sufficient condition. We have added a sentence about it in the last paragraph of the conclusion of the paper, and now also give another example of system (disorder with anisotropic correlated length) where this type of anisotropy is present.

2) I feel slightly uncomfortable in directly transposing the Green function of an isolated cylinder to an ensemble of many cylinders, in particular for what concerns the linewidth of the Mie resonances. In the case of an ordered array, the linewidth in fact disappears and polariton modes with infinite lifetime are formed upon mixing the light and matter resonances. I guess that this does not happen in a disordered medium where scattering processes can instead occur and wavevector is no longer a good quantum number. 
 may anticipate that naive readers like me may appreciate some discussion along these lines.

Response: The fundamental reason why the self-energy of N tubes reduces to N times that of a single tube is that we average over tube positions that are completely randomly distributed in space. This is justified by the fact that the distance between tubes is assumed large compared to their radius, see our reply to point 11) below for more details. In the new version of the manuscript, the steps leading to this conclusion are now explicitly given in the new paragraph around Eqs. (16) and (17). In particular, the new Eq. (17) shows that the  ‘structure factor’ of the array becomes completely uniform upon averaging over the tube positions. In the presence of correlations between tubes, and in particular in the extreme case of a fully ordered array, this would no longer be true and summing over the tubes would make the resonance properties of the array drastically different from those of a single tube, as the referee correctly points out.

3) before eq.(35): the authors choose as a benchmark for all length scales along $z$ the TE scattering mean free path $\ell_\text{scat}$, calculated including the 0 resonance only and taken at the operating frequency \omega. Since the SHE occurs in the vicinity of the 1 resonance, I would have found more natural to use some other length scale that specifically includes also the 1 resonance. Or there is some general argument to justify that $\ell_\text{scat}$ (or better $\ell_\text{scat}^0$) is a valid choice in all cases ?

Response: In fact, the choice of the TE mean free path, which only includes the 0 resonance, as a benchmark for length scales, is the only one that is consistent at all frequency, including the vicinity of the 1 resonance. This statement was justified in the footnote on page 11, but we agree with the referee that it was not enough clearly explained. We have therefore rewritten this note to make the argument as clear as possible. The main point is that in the vicinity of the 1 resonance, the coherent intensity, whose general definition is given by Eq. (38), is still proportional to $\exp(L \text{Im}\Sigma_\text{TE})$ because $|\text{Im} \Sigma_\text{TM} |$ is very large compared to $| \text{Im} \Sigma_\text{TE} |$.

4) along similar lines, I find the authors' choice for x-axis of fig.4 a bit unfortunate, in particular for what concerns the different locations of the narrow peak on the different curves. From what the authors say in the text, I understand that this is not due to a real motion of the resonance, but to the angle-dependence of the reference point $\omega_0$. I suggest the authors to adopt in the whole manuscript a more transparent notation to indicate the frequencies.

Response: We thank the referee for their suggestion. We have replotted the intensity curves in Fig. 4 as a function of $\Delta/\Delta_{10}$, similarly to the convention chosen in the subsequent figures 5, 6 and 7, thus unifying the frequency notation throughout the manuscript. In this representation, one gets rid of the angular dependence of $\omega_0$ and $\Gamma_0$ which was artificially shifting the curves at different $\theta$ with respect to each other in the previous version of the manuscript. Correspondingly, we have simplified the discussion of this figure in the paragraph after Eq. (38).

5) I suspect some problem with the y-axis label of fig.5(a). I would have expected that all $\ell$'s be positive quantities. The caption also contains a suspicious statement "the ratios exceed unity in the vicinity of the resonance" which I can not visualize on the curves.

Response: While the scattering mean free path $\ell_\text{scat}$ is by construction always positive (it is defined defined as the opposite of the imaginary part of the self-energy, i.e., it corresponds to an extinction coefficient), the spatial scales $\ell_\text{S}$ and $\ell_\text{L}$ can have an arbitrary sign because they are defined as self-energy differences. This explains why, in Fig. 5(a), the y-axis is not always positive. This was indeed not enough explained in the manuscript, so we have added a sentence right after Eq. (46) to discuss the sign of $\ell_\text{S}$ and $\ell_\text{L}$. Concerning the statement in the caption of the figure, it was meant to point out that the absolute value of the ratio can exceed unity. We have slightly modified the caption so that the new formulation is clear. In addition, we have added a shaded region in the plot of Fig. 5(a) that explicitly indicates to the reader the range of frequencies where |\ell_scat/\ell_S|>1.

6) eq.(21): I don't find the definition of $\bar{G}(k_\perp,z=L)$. The authors should add it in explicit form somewhere.

Response: We have followed the referee’s suggestion and now give the definition of $\bar{G}(k_\perp,z=L)$ (as a Fourier transform of $\bar{G}(k_\perp,k_z)$) as soon as it is introduced, namely right after Eq. (23).

7) on page.17, it could be worth adding some physical interpretation to the formulas for the time scales of the coherent flash, $\Gamma_0^{-1} \ell_\text{scat}/L$ and $\Gamma_1^{-1} \ell_\text{scat}/L$. In particular, the inverse proportionality to the sample length L is very counterintuitive and deserves justification.

Response: We thank the referee for their suggestion. In the forward direction, the decrease of the emission lifetime by a factor $\ell_\text{scat}/L$ is related to the cooperative nature of the flash effect: at a given distance from the input interface, the scatterers are not only excited by the incoming laser but by a superposition of the incoming laser and the field radiated by the dipoles lying in shallower layers. This leads to an effective reduction of the coherence of the emitted field, all the more important as the medium is thicker. At the end of page 18, we have added a paragraph to physically explain this phenomenon.

8) in the conclusions, I do not understand what the authors mean by "account for the spin-orbit corrections in the multiple-scattering signal". I thought that SO-coupling was already included in the present work. Or am I missing something important?

Response: In our work we consider the impact of spin-orbit coupling on the coherent mode. The latter is only one part of the multiple scattering signal, precisely the part that propagates in the same direction as the incident laser (the ‘forward direction’) and is coherent with it. By the term ‘multiple-scattering’ signal written in the conclusion, we were referring to the photons scattered in other directions. This was indeed slightly misleading and therefore we have completely reformulated this sentence. We have also reformulated the beginning of Sec. 3.2 to more clearly define the coherent mode.

9) The authors should complete the introduction and the bibliography with some discussion of earlier works on the use of Mie resonances to realize photonic crystals and/or metamaterials with exotic properties. I have heard about such developments at several conferences and I am sure there is abundant literature on the topic. These revisions are essential to put the work in a proper context.

Response: We thank the referee for their suggestion. We have followed it and have extended the introduction (top of page 2) to discuss more the concept of Mie resonances, and to mention previous works exploiting them in photonic crystals and dielectric metamaterials. Correspondingly, we have added the new references [36-40].

10) As far as I understand, the authors are assuming that the cylinders are located at random positions but have identical radii. I expect that any spread in the cylinder radius will result in a inhomogeneous broadening of the narrow Mie resonance, a quick decoherence of the flash of light signal, and, finally, to a significant reduction of the SHE magnitude. Most likely, this is the most serious experimental difficulty of the authors' proposal. Even though this is a theoretical proposal, for the sake of completeness it is important that the authors provide some estimate of the robustness of their predictions against a finite spread in the cylinder radii and the discussion of the actual experimental feasibility.

Response: The referee is absolutely right. A dispersion of tube radii is likely to be the main limitation to the reduction of the spin Hall mean free path that we describe near the TM resonance. We have followed the referee’s advice and have added the new section 4.5. to analyze this effect. The results of this analysis are summarized in the new Table 1, which aims to provide practical criteria in view of a possible experimental measurement. Essentially, we find that the ratio $\ell_\text{scat}/\ell_S$ is decreased as soon as the relative dispersion of radii exceeds the relative bandwidth of the TM resonance. Nevertheless, as long as the dispersion remains small compared to the relative bandwidth of the lowest Mie resonance, the decrease remains moderate. Finally, in the conclusion of the manuscript (page 20) we now discuss the experimental feasibility based on a state-of-the-art experimental platform where the SHE should be observed (a glass array fabricated using femto-second writing beams).

11) On page.7, the authors present in a quite rapid way the assumptions they are doing on the statistics of the cylinder positions. What does "uniformly distributed mean" ? How is this condition compatible with the fact that two cylinders can not overlap? What would be the next corrections to this simplified calculations?

Response: We thank the referee for this very relevant question. The assumption of a uniform distribution for the cylinder positions is of course an idealized limit, which requires that the spatial correlations between tubes (related to the fact that they cannot overlap) are negligible. In practice, this condition is well satisfied in our case, where the distance between tubes is typically large compared to their radius. To better explain this assumption and the compatibility with our hypotheses, we have added a new paragraph on page 8 (see, in particular, the new equations (16) and (17) that better justify the assumption of uniform distribution). At leading order, the inclusion of correlations between the tube should involve corrections in $n\rho^2\ll1$. The general treatment of correlated materials, however, is beyond the scope of our work. For interested readers, we have nevertheless added the relevant reference [46] where this problem has been recently reviewed.

12) How necessary is to have a disordered medium to observed the SHE. Would an ordered lattice of cylinders be sufficient to observe a similar or disorder is essential? I think that non-expert readers would appreciate having a bit of discussion in this sense in the introduction.

Response: Disorder is not necessary for observing a spin Hall effect of light, which is a rather general phenomenon occurring in inhomogeneous materials. In particular, similar optical shifts are also  being explored in ordered structures, such as photonic crystals or metasurfaces. Following the referee’s suggestion, at the end of the first paragraph of the introduction we have added a short discussion of these systems, and added the new references [23-29]. In fact, the spirit of our work is rather to show that a spin Hall effect of light can arise even in a disordered system, where one might have naively expect the random nature of the medium to destroy any such effect. To us, the existence of spin-orbit interactions after ensemble averaging was not obvious in the first place.

---

## Round 1 · Referee Report · Anonymous (Referee 2) · 2022-12-15

Report

The authors have studied the spin Hall effect (SHE) of light in disordered dielectric tubes in the resonant regime. Their study is mainly analytical, supplemented with numerics to show the behavior of various parameters. The detailed analytical treatment comes in a cost that the manuscript is quite technical, but this is understandable and reasonable.

The calculations are sound and the manuscript is well written. The flow of the paper is also good and the details between different steps are explained reasonably. To me, the manuscript can eventually be published in SciPost Physics. However, before I can recommend it for publication, it would be nice if the authors can address and comments on the following points:

Requested changes

  1. What is the prospect of observing this SHE of light in a state-of-the-art experiment? The authors should clarify this in the manuscript and give some prospects.

  2. Regarding spin-orbit coupling (SOC), for charged particles there exist various forms of SOC (although they all have the same origin), such as the Rashba SOC or the Dresselhaus SOC. And each SOC leads to different physical phenomena. Are there also various forms of SOC for light? If so, would all of them lead to SHE?

  3. Again in charged electronic systems, the quantum Hall effect and quantum SHE have topological origins and are characterized by non-trivial topological invariants -- Chern numbers. The resultant edge (or surface) currents are also protected by symmetry and topology. However, in the current work it's completely unclear that this effect is really analogous to the electronic SHE. For such a strong claim, the authors should really show that the system has a non-trivial topology (e.g., the system is characterized by a non-trivial topological invariant). Otherwise, this claim is too vague.

  4. Forgetting about resonance for a moment, I'm wondering how the results obtained fo a system consisting of disordered tubes as in the current work can approach to the results obtained for a continuum disordered medium as in the literature in the appropriate limit?

  5. Comparing Figs. 5(b) and 6(a), it's not clear why some $R_y^\delta(L)$ curves in 6(a) go back to zero in large $L$, while in 5(b) they are all nonzero and saturate for large $L$.

  • validity: good
  • significance: good
  • originality: good
  • clarity: high
  • formatting: perfect
  • grammar: perfect

Author:  Nicolas Cherroret  on 2023-01-16  [id 3241]

(in reply to Report 2 on 2022-12-15)
Category:
answer to question

We thank the referee for recommending our paper for publication. We address their comments and questions hereafter.

  1. What is the prospect of observing this SHE of light in a state-of-the-art experiment? The authors should clarify this in the manuscript and give some prospects.

Response: We thank the referee for their question and suggestion. We agree that a discussion of the experimental feasibility was missing in the manuscript. In the new version, we have addressed it by adding a paragraph in the conclusion on page 20, where we discuss a state-of-the-art experimental platform where the SHE should be observed (a glass array fabricated using femto-second writing beams) and explain how our results apply to that system. Furthermore, in the new version of the manuscript we have added the new section 4.5, where the impact of the dispersity in tube radii on the SHE is theoretically analyzed. This dispersity, indeed, is presumably the main limitation to an experimental observation of the SHE in a random medium.

2. Regarding spin-orbit coupling (SOC), for charged particles there exist various forms of SOC (although they all have the same origin), such as the Rashba SOC or the Dresselhaus SOC. And each SOC leads to different physical phenomena. Are there also various forms of SOC for light? If so, would all of them lead to SHE?

Response: The situation in optics is similar to that for charged particles mentioned by the referee. A common feature of all forms of spin-orbit coupling is that they originate from the coupling between the spatial variation of the electric field and its direction. However, depending on the material/system under consideration, SOC can manifest itself in different fashions. Although, to our knowledge, a rigorous classification is not firmly established, at least three forms of optical SOC have been identified:  SOC in inhomogeneous isotropic materials such as interfaces, SOC in free-space but in strongly focused beams, and SOC in inhomogeneous anisotropic materials (to which our disordered system somehow belongs). Concerning the SHE, it is a more specific phenomenon than SOC but, in general, as soon as SOC is present a SHE can be observed (with possibly different properties).

  1. Again in charged electronic systems, the quantum Hall effect and quantum SHE have topological origins and are characterized by non-trivial topological invariants - Chern numbers. The resultant edge (or surface) currents are also protected by symmetry and topology. However, in the current work it's completely unclear that this effect is really analogous to the electronic SHE. For such a strong claim, the authors should really show that the system has a non-trivial topology (e.g., the system is characterized by a non-trivial topological invariant). Otherwise, this claim is too vague.

Response: In the present work, we describe an optical analogue of the simple spin Hall effect, which is a phenomenon that only involves the semi-classical motion of electrons in the presence of spin-orbit coupling. On the other hand, we do not make any claim about an analogy with the quantum spin Hall effect which, as the referee correctly points out, is related to protected edge states. At the end of the conclusion, we have added a sentence mentioning the quantum spin Hall effect, together with two relevant references (Refs. 52 and 53 in the new version of the manuscript) where optical analogues of this phenomenon have been studied. The question of an optical quantum spin Hall effect in disordered media has, on the other hand, not yet been addressed to our knowledge, but is an interesting question for future work.

4. Forgetting about resonance for a moment, I'm wondering how the results obtained for a system consisting of disordered tubes as in the current work can approach to the results obtained for a continuum disordered medium as in the literature in the appropriate limit?

Response: The case of continuous disorder (where no resonances arise) is recovered in the limit of vanishing frequency (precisely, for a wavelength very large compared to the tube radius). This limit is discussed in the paragraph around Eq. (50) of the new version of the manuscript. To better emphasize this point, on page 14 we have changed the sentence ‘Let us first consider the static limit […] which corresponds to the model of non-resonant dielectric media […]’ to  ‘Let us first consider the static limit […] which corresponds to the model of non-resonant, continuous disordered media […]’.

5. Comparing Figs. 5(b) and 6(a), it's not clear why some $R_y^\delta(y)$ curves in 6(a) go back to zero in large L, while in 5(b) they are all nonzero and saturate for large L.

Response: Generally speaking, the spin Hall shifts in Fig. 5(b) and 6(a) have no obvious connection. Indeed, in the first case the spin Hall effect corresponds to a global shift of the whole beam, while in the second case the polarimetric detection post-selects a sub-part of the whole beam. A mathematical consequence of this different procedure, pointed out in Ref. [31], is that in the polarimetric measurement the shift depends on the sign of the spin Hall mean free path $\ell_\text{S}$, which directly impacts the behavior of $R_y^\delta$ at large $L$.  n particular, near the TM resonance $\ell_\text{S}$ is negative which leads to a decay of the shift with L. Away from the TM resonance, however, the sign of $\ell_\text{S}$ changes, which would yield a saturation of the shift.

---

## Round 2 · Referee Report · Anonymous (Referee 3) · 2023-2-3

Report

I appreciate the authors' efforts to revise the manuscript in response to my and the other Referee's remarks. I am (almost) happy to recommend publication of the revised manuscript.

Before that, the authors may consider the following (optional) suggestions:

-I understand that a discussion on the role of broken symmetries to induced the SHE goes beyond this work. Still, any reinforment of the manuscript in this sense would be most welcome and of utility for the reader.

-In the newly added discussion on femto-second laser-written waveguide arrays, the authors correctly mention that in these systems the refractive index contrast m is always very close to 1. Since this regime appears to be very different from the one considered in the figures (where m=5 or m=3 was taken), it may be useful to spend some words in discussing how Mie resonances change when m->1 (for instance I expect their linewidth to get much broader and their contrast to be suppressed) and, then, what is the lower bound on |m-1| to observe a SHE effect (for m=1 the effect should completely disappear).

Requested changes

see above for my suggestions

---

## Round 2 · Referee Report · Anonymous (Referee 4) · 2023-2-6

Report

Authors have addressed my questions and remarks satisfactorily. Therefore, I believe the paper can be published in SciPost.

---

## Round 2 · Author Response

Dear Editor,

We would like to resubmit our manuscript entitled "Resonant Spin Hall Effect of Light in Random Photonic Arrays" to SciPost Physics. The manuscript has been previously reviewed by two anonymous referees, who both recommended it for publication. They also asked a number of relevant scientific questions and made a few suggestions to improve the manuscript. We have addressed all these points in separate replies to the referees. Accordingly, we have made a few changes in the manuscript, which are listed below. In the preprint file enclosed, the main changes also appear in red, so that they can easily be identified. Together we our reply to the referees, we hope that these changes make our work publishable in SciPost Physics.

Yours sincerely.

The authors.

---

## Round 2 · List of Changes

1- We have extended the paragraphs around Eq. (43) and added a sentence in the second paragraph of the introduction to better discuss the role of the initial polarization.

2- We have added a sentence about the symmetry properties of the disorder in the last paragraph of the conclusion of the paper.

3- The steps and conditions allowing us to derive the self-energy of an ensemble of many cylinders from the t-matrix of an isolated cylinder are now explained in details in the new paragraph around Eqs. (16) and (17). We have also added the new reference [46] to refer the interested reader to the case of correlated or partially disordered media.

4- On page 11, we have slightly modified the footnote 1 to better explain our definition of the mean free path.

5- We have replotted Fig. 4 as a function of \Delta/\Delta_10, and have slightly shortened the discussion of this figure in the paragraph following Eq. (38).

6- We have added a sentence right after Eq. (46) to discuss the sign of \ell_S and \ell_L,

7- We have slightly modified the caption of Fig. 5 and have added a shaded region in the plot of Fig. 5(a).

8- We have added the definition of \bar{G}(k_\perp,z=L) right after Eq. (23).

9- At the end of page 19, we have added a paragraph to qualitatively explain the physical origin of the time scale of the coherent flash.

10- We have reformulated the discussion on multiple scattering in the last paragraph of the conclusion.

11- We have reformulated the beginning of Sec. 3.2, so that the definition of the coherent mode is clearly given.

12- On page 15 we have added the new section 4.5 to discuss the impact of a dispersion of tube radii.

13- On page 20, in the conclusion: we have added a paragraph to discuss a state-of-the-art experimental platform where the SHE should be observed.

14- We have added a new paragraph on pages 8 to better explain our assumption of uniform distribution for the scatterer positions, as well as the new reference [46].

15- At the end of the first paragraph of the introduction, we have added a discussion on spin Hall effect in photonic crystals and metasurfaces, together with the new references [23-29].

16- At the end of the conclusion, we have added a sentence mentioning the quantum spin Hall effect, together with the two new references 52 and 53.

17- On page 14, we have changed ‘ […] which corresponds to the model of non-resonant dielectric media […]’ to  ‘ […] which corresponds to the model of non-resonant, continuous disordered media […]’.

18- In Eq. (51), we have added the explicit expression of the scattering mean free path near the TM resonance.

19. We have slightly amended the introduction of the manuscript.

---

## Editorial Decision

published